



**Europe's adaptation to the energy crisis: Reshaped gas supply-**
**transmission-consumption structures and driving factors from**
**2022 to 2024**
Chuanlong Zhou[1], Biqing Zhu[1], Antoine Halff[2,5], Steven J. Davis[3], Zhu Liu[4], Simon Bowring[1], Simon Ben
Arous[5], Philippe Ciais[1,*]
[1] Laboratoire des Sciences du Climat et de l'Environnement, CEA CNRS UVSQ, 91190, France
[2] SIPA Center on Global Energy Policy, Columbia University, New York, NY 10027, USA
[3] Department of Earth System Science, University of California Irvine, Irvine, CA 92697, USA
[4] Department of Earth System Science, Tsinghua University, Beijing, 100190, China
[5] Kayrros Inc., Paris, 75009, France
*Correspondence to*: Philippe Ciais.(philippe.ciais@lsce.ipsl.fr)
**Keywords:** natural gas, EU energy crisis, supply-transmission-consumption analysis, LNG
**Abstract.** The 2022 invasion of Ukraine by Russia triggered a significant energy crisis in the EU27&UK, leading to
profound changes in their natural gas supply, transmission, and consumption dynamics. To analyze those pattern
shifts, we first update our natural gas supply dataset, EUGasSC, with daily country- and sector-specific supply
sources. We then provide a newly constructed daily intra-EU natural gas transmission dataset, EUGasNet, with
specified supply sources utilizing the ENTSOG (European Network of Transmission System Operators for Gas) and
EUGaSC data. To further understand the economic and climatic impacts, we finally developed EUGasImpact, a
daily dataset with sector-specific driving factors of consumption changes based on change attribution models using
multiple open datasets. Those datasets are available on the Zenodo platform:
https://doi.org/10.5281/zenodo.11175364 (Zhou et al., 2024). On the supply side, Russian gas supply to the
EU27&UK was cut by **87.8%** (**976.8 TWh per winter**) during the post-invasion winters compared to the previous
winters. LNG imports become the largest gas supply source, rising from **20.7**% to **37.5**% of the total gas supply. Our
intra-EU gas transmission analysis showed the gas transmission network was adjusted to mitigate the large gas
shortfalls in Germany and distribute LNG arrivals. Total gas consumption fell by **19.0%**, which was driven by 1)
consumer behavioral changes in household heating (contributed to **28.5%** of the total reduction, the same for the
following numbers), 2) drops in industrial production (**24.5%**), 3) heating drops due to the warmer winter
temperatures (**10.6%**), 4) shifts towards renewable electricity including wind, solar, and hydro (**10.2%**), 5) decline
in gas-powered electricity generation (**9.4%**), 6) adoptions of energy-efficient heat pumps for industrial gas heating
(**4.2%**), 7) shifts towards non-renewable electricity including coal, oil, and nuclear (**0.8%**), and 8) other unmodeled
factors (**11.8%**). We evaluated the benefits and costs associated with these pattern changes and discussed whether
these changes would potentially lead to long-term structural changes in the EU energy dynamics. Our datasets and
these insights can provide valuable perspectives for understanding the consequences of this energy crisis and the
challenges to future energy security in the EU.





## 1.Introduction

The European Union faced an energy crisis triggered by the Russian invasion of Ukraine in 2022, which led to a sudden halt of natural gas supplies from Russia, the EU's largest gas supplier (Conscious uncoupling: Europeans' Russian gas challenge in 2023, 2023). In the year prior, 2021, Russia exported $155 \times 10^9$ m3 of natural gas ($1.6 \times 10^3$ TWh), accounting for 45% of the total EU gas imports (Eurostat, n.d.). The subsequent winters of this crisis presented crucial tests for the EU's ability to manage the disruption in Russian pipeline gas imports (A Test Of Endurance: Europe Faces A Chilling Couple Of Years, But Russia Stands To Lose The Energy Showdown, 2023), especially considering the high heating demand in the cold seasons and the unexpected energy supply shortfalls (Zhou et al., 2023).

Prior studies have suggested multiple strategies to mitigate the energy crisis: 1) on the supply side, increasing gas imports from alternative suppliers, including additional imports from LNG and existing pipelines from Northern Africa and Middle East (Preparing for the next winter: Europe's gas outlook for 2023, 2023) (The gas situation in Europe remains favorable thanks to contained demand and stable supply, resulting in g…, 2023), 2) in terms of gas transmission, enhancing the intra-EU gas redistribution to balance the supply and demand among member states (Zhou et al., 2023), 3) on the consumption side, reducing gas demand, such as energy conservation, scaling down industrial production (Baseline European Union gas demand and supply in 2023 – How to Avoid Gas Shortages in the European Union in 2023 – Analysis, 2023) (Europe's energy crisis: What factors drove the record fall in natural gas demand in 2022?, 2023), 4) regarding the energy structure, diversifying energy sources from gas, for instance, increasing the number of heat pumps, and switching to non-gas-powered electricity (Conscious uncoupling: Europeans' Russian gas challenge in 2023, 2023) (Averting Crisis, Europe Learns to Live Without Russian Energy, 2023). However, there is a pressing need for detailed, high-resolution data to quantify the significance of those pathways in structural supply-transmission-consumption pattern changes and their economic-climatic impacts after the energy crisis. Additionally, quantitative assessment of intra-EU gas redirection in mitigating the crisis remains unclear, particularly since there were transmission "bottlenecks" in the intra-EU gas transmission net (Zhou et al., 2023) (LNG exports for selected countries, 2015-2025, 2020).

To address these needs, we developed comprehensive datasets for supply, transmission, and consumption at daily resolution, respectively. On the supply side, we updated our EU natural gas dataset, EUGasSC (Zhou et al., 2023), which provides the country- and sector-specific gas supply patterns. For analyzing intra-EU gas redirection, we provide a newly constructed intra-EU gas transmission net dataset, EUGasNet, utilizing the ENTSOG (European Network of Transmission System Operators for Gas) (ENTSOG, 2023) and EUGasSC data. To quantify the driving factors in the heating, power, and industrial sectors, we developed EUGasImpact using sectoral-specific change attribution models and multiple open datasets, which enables the factor analysis and impact analysis of consumption changes across these sectors.

For the heating sector, we differentiated the consumption changes with the contribution of consumers' behavioral changes and anomalously warm winter temperatures (Abnett, 2023) based on a behavior-climate attribution model utilizing temperature-consumption curves (Zhou et al., 2023) and ERA5-land temperature data (Hersbach et al., n.d.).

For the power sector, we assessed whether the consumption reduction in gas-powered electricity generation could be



offset by the increments of electricity generation from alternative energy sources. This evaluation was conducted through an explainable-reduction attribution model with day-to-day change comparisons, utilizing both gas consumption and electricity generation data from the Carbon Monitor power dataset (Zhu et al., 2023). For the industrial sector, we assume the consumption reduction due to the substitution of gas heating with electrically powered heat pumps is unlikely to lead to declines in industrial production (The gas situation in Europe remains favorable thanks to contained demand and stable supply, resulting in g…, 2023). Similar approach is performed to assess whether the industrial consumption reduction could be explained by the increments of total electricity generation with certain conversion efficiency.

We provide three datasets, EUGasSC, EUGasNet, and EUGasImpact, which provide daily gas supply, transmission, and consumption dynamics with country- and sector-specific data. Using these datasets, we conducted comprehensive analyses of how EU states adapted to the energy crisis triggered by the invasion. The EUGasSC dataset illustrates how LNG imports replaced Russian pipeline imports and become the primary gas supply source to the EU. With EUGasNet, we found intra-EU gas transmission network shows enhanced "redirection" flows towards the western states, especially to Germany, which faced the most acute gas shortage. According to EUGasImpact, we quantified the contributions of driving factors to the significant net gas consumption reduction during the post-invasion winters, such as heating reduction, energy structure shifting, structural dependency reduction of gas use, and declines in electricity generation and industrial production. We further discussed the economic and climatic consequences of those shifts found from the three datasets, as well as whether those facets would lead to structural transformations in long-term energy security within the EU regions, regarding the uncertainties associated with the global LNG market, the intra-EU transmission "bottleneck" in the EU gas network, potential impact to residential living costs, and the ongoing transition towards greener energy sources.

## 2.Methods

### 2.1.Data collection

The workflow of this study is shown in Fig 1. On the supply side, we collected the EUGasSC dataset that provides the daily country- and sector-specific gas supply data (Zhou et al., 2023). For the analysis of intra-EU gas transmission, we gathered the daily natural gas transmission (pipeline) and import data (both pipeline and LNG imports) for the EU27&UK from ENTSOG (ENTSOG, 2023), and used EUGasSC for specifying the supply sources.

The ERA5-land temperature (Hersbach et al., n.d.) was collected and used to fit the empirical temperature-gas-consumption (TGC) curves to estimate the gas consumption changes in the heating sector (Zhou et al., 2023) (Ciais et al., 2022). The country-based daily power generation data with specified energy sources, including gas, coal, oil, nuclear, wind, solar, hydro, and other renewables, were collected from the Carbon Monitor power dataset (Zhu et al., 2023).

The Dutch Title Transfer Facility (TTF) natural gas prices from 2019 to 2024 were collected and used as the overall natural gas price index in EU27&UK (Dutch TTF Natural Gas Futures, 2023). The household energy price index



(HEPI) and the gas and electricity prices in the capital cities of EU27&UK were used for the analysis of economic
impacts on household gas consumption (Household Energy Price Index, 2023).

### 2.2. Periods of analysis

The "winter" in this study refers to the major heating months that are associated with an elevated risk due to the energy
shortage. Therefore, "winter" is defined to include the months of November, December, January, February, and March.
These months use gas intensively as a major heating fuel (contributing from 54.3% to 58.8% of annual gas
consumption) in the EU countries (Zhou et al., 2023). Accordingly, we define the post-invasion winters as November
2022 to March 2023 for the winter of 2022-2023, and November 2023 to March 2024 for the winter of 2023-2024.
For comparative analysis, we refer to the three pre-invasion winters of 2019-2020, 2020-2021, and 2021-2022, using
the same seasonal timeframe.
However, our study encompasses an annual perspective for analyzing the net intra-EU gas transmission. The intra-EU
gas transmission in the non-heating seasons can also be important as they may indicate the variations and
redistributions of gas storage within the EU (Zhou et al., 2023). Therefore, the pre-invasion period is defined from
2019-04-01 to 2022-03-31 (three years), and the post-invasion period is defined from 2022-04-01 to 2024-03-31, as
illustrated in Fig 3.

### 2.3. EU gas supply (EUGasSC)

For the supply side, we updated our EU natural gas dataset, EUGasSC, extending its coverage until 2024-03-31. The
EUGasSC dataset has been described in our previous work (Zhou et al., 2023) and briefly introduced in the
supplementary. The EUGasSC dataset provides the daily country- and sector-specific gas supply based on a mass flow
balance simulation model. We then estimated changes in gas supply sources, including Russian imports, LNG imports,
other pipeline imports, and EU local productions based on the EUGasSC dataset. We observed a supply shortfall, the
"Russian gas gap", for post-invasion winters due to the inability to boost non-Russian gas supplies to offset the
reduction in Russian gas supply. However, this gap in gas supply did not necessarily translate into a "shortage". This
supply-consumption dynamic analysis will be discussed below in section 2.5.
Note that the gas supply discussed in this paper refers to the original supply source estimated in EUGasSC dataset
(Zhou et al., 2023) For example, Germany may receive LNG gas supplies even though there are no LNG terminals
in Germany before Dec. 2022 (Ukraine war pushes Germany to build LNG terminals, 2023).

### 2.4. Intra-EU gas transmission (EUGasNet)

To understand the changes in the intra-EU gas transmission in response to the energy crisis, we analyzed the net flow
changes in the gas transmission network between the pre-invasion and post-invasion periods. To perform the net flow
change analysis, we first constructed the gas transmission network graphs by integrating the physical flows, import
volumes from ENTSOG, and supply source from EUGasSC for both pre-invasion and post-invasion periods (Fig S7
b and c). Then we access the the bidirectional flow differences between the annual average transmission values of the



pre- and post-invasion periods (Fig S7 a). Finally, we computed the net flow changes by accumulating the bidirectional
flow differences. The detailed equations are presented in the supplementary.
This net flow change analysis allows us to understand the shifts in significance of both countries (nodes) and their
interconnections (edges) in the intra-EU gas transmission network, as shown in Fig 3. The nodes are color-coded to
represent countries experiencing either an increase (in green) or a decrease (in red) in outgoing gas transmission
relative to the pre-invasion period. The direction and net flow change (edges between countries) are only meaningful
if analyzed together. For example, a positive edge connected from France to Germany indicates an increased net flow
from France to Germany relative to the pre-invasion period (Fig S8), and this is equivalent to a negative edge from
Germany to France. The edge directions in our analysis (Fig 3) are defined based on the flow patterns observed in the
pre-invasion network. Therefore, the red edges in Fig 3 indicate reversed transmission directions between the two
countries during the post-invasion periods.

## 150    2.5.Consumption changes (EUGasImpact)

The EU27&UK responded to the "Russian gas gap" during the post-invasion winters by diversifying gas supplies,
conserving usage, and reducing structural gas dependency. To further understand those dynamics and their impacts,
we developed consumption reduction attribution models for residential heating, power, and industrial sectors (Fig 1).
EUGasImpact is then constructed based on the output of these reduction attribution models at daily resolution. The
detailed model equations for all the sectors discussed below are presented in the supplementary.

### 156    2.5.1.Residential heating sector

In the residential heating sector, we assess the impact of heating behavioral changes and climate change based on the
behavior-climate attribution model (Fig 1). This approach utilized the empirical Temperature-Gas Consumption (TGC)
curves, which illustrate how heating consumption varies with changes in ambient temperature (Zhou et al., 2023)
(Ciais et al., 2022). We developed the TGC curves for both pre- and post-invasion winters to capture the shifts in
residential heating behaviors (Fig S9). The gas consumption change due to the behavioral shifts can be estimated by
calculating the differences in consumptions at post-invasion temperatures using both pre-invasion and post-invasion
TGC curves. Similarly, gas consumption changes due to temperature variations can be estimated by computing the
differences in consumption under pre-invasion and post-invasion temperatures using the post-invasion TGC curves.

### 165    2.5.2.Power sector

In the power sector, we assess whether the reduction in gas consumption for electricity generation can be offset by
alternative sources (if exist), or lead to a net decrease in electricity supply based on the explainable-reduction
attribution model (Fig 1). We assume that any reduction in gas-powered electricity could be compensated by increased
electricity generation from coal, oil, nuclear, wind, solar, hydro, and other forms. Conversely, an inability to fill the
reduction in gas-powered electricity might suggest a potential shortage in the overall electricity supply. To smooth
out weekly variations, we utilized 7-day aggregated data for day-to-day comparisons of all energy sources during both
the pre- and post-invasion winters.



### 2.5.3.Industrial sector

In the industrial sector, gas consumption can be differentiated between gas consumption for energy use, such as heating and electricity generation, and non-energy use, like chemical feedstocks or raw materials (Supply, transformation and consumption of gas, 2023) (Energy Statistics, 2023). Therefore, gas consumption reduction resulting from the adoption of heat pumps is unlikely to negatively impact industrial production, however, reductions in non-energy gas use may indicate a decline in industrial output  (The gas situation in Europe remains favorable thanks to contained demand and stable supply, resulting in g…, 2023) (Preparing for the next winter: Europe's gas outlook for 2023, 2023). Like the power sector, we evaluate the potential impact of reduced gas consumption on industrial production using the explainable-reduction attribution model (Fig 1) and 7-day aggregated comparisons. We assume that any increase in electricity generation is primarily due to heightened heat pump usage in industry, resulting in lower gas consumption for energy use. A decrease in industrial gas consumption is unlikely to negatively affect industrial production if the increase in electricity generation (if present) is sufficient to compensate for the reduced gas use, considering a specific gas-to-electricity conversion efficiency (Fig S16).

### 3.Uncertainties and bias

In the residential heating sector, uncertainties are relatively low as TGC curves can effectively capture the gas consumption based on temperature (Fig S9, r2 = 0.55±0.21). The estimated consumption changes ( $change_{behavior,date}$ + $change_{temperature,date}$ ) account for 94.0±13.2% of the actual changes ($consumption_{pre\_date}$ - $consumption_{post\_date}$). This low model uncertainty also underpins the precise predictions of gas conservation in our previous study, with only a slight overestimation of 4.6% (Fig 6, right panel).

In the power and industrial sectors, our attribution model assumed constant total power generation volumes and electricity demands during the pre- and post-invasion winters. The differences in total power generation between pre- and post-invasion winters were relatively small (-0.4±0.6%) across the EU27&UK. However, our assumption of unchanged electricity demand could lead to an overestimation of "negative impacts",  i.e., power supply shortages or negative effects on industrial production. This simplification overlooks demand variations in response to high electricity prices and energy conservation measures in EU countries (What measures are European countries taking to conserve energy?, 2023) (Averting Crisis, Europe Learns to Live Without Russian Energy, 2023). Additionally, we did not account for the interconnection of the EU power grid, which could help balance differences in electricity supply and demand at the country level. As a result, our analysis might depict the "maximum" potential negative impacts of gas reductions on the power and industrial sectors.

### 4.Results

### 4.1.Overview of gas supply and consumption

During the post-invasion winters, the natural gas supply structure to EU27&UK was profoundly reshaped (Fig 2 and Fig S1). The share of EU gas supply from Russia, the previous largest supplier, plummeted from 36.3% to 5.4%,



creating a shortfall of 976.8 TWh per winter. Despite this dramatic reduction, Russia continued to provide a
considerable volume of gas (257.3 TWh) to EU27&UK during the post-invasion winters through the ongoing
transmissions to Slovakia, Lithuania, Poland, and Hungary (Table S1), and the non-winter gas storage (12.9 TWh, Fig
S4). The supply gap from Russia was filled by 43.5% in the two post-invasion winters, primarily through the increased
LNG imports (593.3 TWh), and scaling up pipeline throughput from Norway and Serbia (176.9 TWh), Libya and
Algeria (79.9 TWh). Conversely, gas supplies from Middle Eastern countries (Turkey and Azerbaijan) and EU
production decreased by 13.2 TWh and 45.4 TWh, respectively, during the post-invasion winters. The remaining
Russian gas supply gap, combined with the other supply drops, led to a substantial reduction in gas consumption
during the post-invasion winters, amounting to 581.0 TWh per winter.
We observed a uniform decrease in gas consumption (25.5±16.0%) across all EU countries regardless of their varying
levels of reliance on Russian gas. In Western EU countries (Fig 6 and Fig S3), where the gas supply sources remained
robust, consumption reductions surpassed the decline in the Russian gas supply. This suggests that demand-side
factors, such as higher gas prices or a shift away from structural dependence on gas (Europe's energy crisis: What
factors drove the record fall in natural gas demand in 2022?, 2023), were likely the primary drivers behind these
reductions. In contrast, in other EU countries (Fig 6 and Fig S3), the gaps in Russian gas supply were greater than the
reductions in consumption, indicating that supply-side constraints, such as the lack of sufficient alternative gas sources
to compensate for the reduced Russian supplies, played a more significant role in their consumption reductions.
**4.2. Changes in intra-EU gas transmission**
Following the Russian invasion of Ukraine, significant changes occurred in intra-EU gas transmissions(Fig 3 and Fig
S7). Gas transmissions from Central and Eastern to Western EU countries dominated intra-European gas networks
during the pre-invasion periods. However, these flows experienced sharp declines in both cross-border flows (red lines
in the network, Fig 3) and total country outflow (red circles in the network, Fig 3), primarily due to a substantial
reduction of Russian gas exports to the EU (-2427.5 TWh per year). In response, gas transmission in the opposite
direction, from Western to Eastern EU countries (green lines and circles in the network, Fig 3), increased as a
compensatory measure to the reduced Russian supply. Our flow change analysis (Fig 3, S1, S7, and S8) was conducted
over a one-year duration to fully account for the transient seasonal flows related to storage changes (see method 2.4).
Two critical pathways showing reduced net Russian gas transmission are evident in the network (Fig 3, marked in
red): 1) from Russia, Slovakia, and Austria, to Italy, and 2) from Russia, Poland, to Germany. Notably, a larger
negative net flow from Poland to Germany (-520 TWh per year) compared to Russia to Poland (-373 TWh per year)
appears counterintuitive. The reason for this is the shift in the initial net flow direction: prior to the invasion, gas
flowed from Poland to Germany; but during the post-invasion period, Germany reversed the flow, sending back part
of its gas imports from Western countries to Poland (Fig S8). To compensate for the reduced Russian supply at the
EU scale, the major LNG importing countries, including Spain, the UK, Portugal, the Netherlands, France, and
Belgium-Luxembourg, significantly increased their LNG transmission over consumption ratio from 0.36 to 1.08
(Table S2), indicating that a greater portion of the LNG imported by these countries was redirected to others with
larger gas deficits.



### 4.3. Consumption reductions and attributions

The sectoral gas consumption reductions per winter are ranked in decreasing order as follows: residential sector (208.5 TWh, accounting for 14.1% of the sector) > industrial sector (153.3 TWh, 27.5% of the sector) > power sector (108.6 TWh, 19.5% of the sector). However, these reductions in consumption do not necessarily equate to gas shortages in EU countries, as various responses can either reduce the gas demand or structural gas dependency (Conscious uncoupling: Europeans' Russian gas challenge in 2023, 2023)(The gas situation in Europe remains favorable thanks to contained demand and stable supply, resulting in g…, 2023)(Baseline European Union gas demand and supply in 2023 – How to Avoid Gas Shortages in the European Union in 2023 – Analysis, 2023). Based on our reduction attribution models (Fig 4 to 5), we attributed the gas consumption reductions to the following factors: 1) behavioral/structural change (43.7%), which includes decreased household heating consumption (28.5%), increased electricity supply from alternative energy sources (coal, nuclear, wind, solar, and hydro, 11.0%), and the adoption of heat pumps in the industry (4.2%), 2) gas shortage (33.9%), including declines in electricity generation (9.4%) and industrial production (24.5%), and 3) other factors (22.4%), such as reduced heating demand due to warmer temperatures (10.6%), and changes in unmodeled consumptions (11.8%).

### 4.3.1. Residential heating sector

We first examined the consumption reduction in the residential heating sector as gas is mainly used for heating ($46.2 \pm 18.0\%$ of total consumption for the post-invasion periods). Our findings reveal that, in the post-invasion winters, the majority of EU27&UK countries reduced their consumption in both household and public heating, ranging from -0.5% to -59.3% compared with pre-invasion winters with exceptions in Poland and Finland with increased consumptions. Italy experienced the largest absolute consumption reduction in the heating sector (35.6 TWh, accounting for 16.7% of its consumption), followed by Germany (35.3 TWh, -13.6%), the UK (30.4 TWh, -11.2%), France (28.7 TWh, -16.7%), and Hungary (26.8 TWh, -27.6%). Although warmer countries could have larger reduction potentials in the residential sector, we did not find a significant correlation between heating consumption reduction and mean winter temperature ($p=0.24$, Fig S10).

Conversely, we discovered that reductions in gas consumption within the residential heating sector were positively correlated with temperature anomalies (Pearson's $r= 0.49$, $p<0.05$, Fig 4a). To explore the impact of warmer temperature anomalies, we developed a behavior-climate attribution model using TGC curves that attribute consumption reductions to either behavioral changes or temperature variations (see method). We found lower heating consumptions for the same temperature (flatter slope of TGC diagrams) and a lower heating inception temperature (smaller intercepts of TGC diagrams) during the post-invasion winters. The consumption changes in the residential sector were primarily attributed to behavioral change (72.9%, Fig 4a), even though warmer post-invasion winter conditions have been extensively reported to mitigate the impact of gas supply reductions in the EU (A Test Of Endurance: Europe Faces A Chilling Couple Of Years, But Russia Stands To Lose The Energy Showdown, 2023)(Preparing for the next winter: Europe's gas outlook for 2023, 2023). Those significant behavioral saving changes can be the potential response to intentional reductions due to high energy prices, government campaigns, or



277  structural shifts away from fossil gas use for heating with heating pumps (Preparing for the next winter: Europe's gas
278  outlook for 2023, 2023) (Denmark launches energy saving campaign; European gas supply "under pressure," 2023).

### 4.3.2. Power sector

During the post-invasion winters, all EU27&UK countries decreased their reliance on gas-powered electricity, with reductions ranging from 15.9% to 66.5% compared to pre-invasion levels. To explore whether these reductions might lead to an energy shortage (net power generation decline), we compared the mix of electricity generation sources for the pre- and post-invasion periods based on an explainable-reduction attribution model, which assesses whether the decrease in gas-powered electricity generation could be replaced by other energy sources (see method).

Our analysis reveals that, on average, 35.0±22.9% of the days during the post-invasion winters in the EU27&UK experienced a net reduction in electricity generation due to decreased gas consumption in the power sector. These reductions accounted for 57.0% of the total electricity generation decline during the post-invasion winters (35.2 TWh per winter). In particular, Italy experienced the largest and longest duration of electricity generation drop caused by gas reductions (-7.5 TWh and 77.2% of winter days), followed by the UK(-3.8 TWh and 58.9%), and the Netherlands (-1.7 TWh and 58.3%), as depicted in Fig 4b.

Following the energy structural change in the power generation between the pre-invasion and post-invasion periods (Fig S12 and S13), we estimate that 79.6±47.7% of the gas reduction in the power sector was compensated by alternative power sources (Fig 4b). Among these substitutes, renewables including wind, solar, and hydropower, contributed the majority of that substitution at 80.0%±22.0%, and their increases were always strongly correlated with the deficits of gas-powered electricity (Pearson's r = -0.79, p<0.05, Fig 4b and S15). Countries with the highest share of renewable-power substitution included Spain (9.2 TWh), followed by the Netherlands (7.2 TWh) and France (6.6 TWh). Substitution by oil and coal, contributed only 10.2±13.3%, while nuclear power contributed only a small proportion of 4.5% (Fig 4b) because French reactors had an extremely low availability.

Compared to our initial predictions about how the shortfall in Russian gas would be addressed (Fig 6, right panel), the power sector exhibited the largest discrepancy. We had largely overestimated European gas-substitution capacity in power generation as we did not anticipate the shortfall of nuclear and hydropower in France (Fig S14) (List of outages and messages, 2023) (The gas situation in Europe remains favorable thanks to contained demand and stable supply, resulting in g…, 2023).

### 4.3.3. Industrial sector

We lastly looked at adjustments to industrial production, a sector particularly sensitive to supply-side energy shocks or high gas prices. All EU27&UK countries reduced their industrial gas consumption during the post-invasion winters by an average of 26.3±14.6%. Germany saw the largest reduction per winter (41.0 TWh), contributing 25.1% of the total reduction across EU27&UK, followed by Spain (31.6 TWh), France (14.2 TWh), and Italy (12.3 TWh).

The reduction in industrial gas consumption, both for energy and raw material uses, can lead to a decline in industrial output (Supply, transformation and consumption of gas, 2023)(Energy Statistics, 2023). However, the industrial gas consumption reduction in energy use can also be associated with a structural adjustment to heating techniques, such



as the adoption of heat pumps  (see method and Fig S16), which were not expected to negatively impact industrial
production (Preparing for the next winter: Europe's gas outlook for 2023, 2023)(The gas situation in Europe remains
favorable thanks to contained demand and stable supply, resulting in g…, 2023).
Using a gas-electricity conversion efficiency of 0.7, our reduction attribution model indicates that on 69.5% of the
post-invasion winter days (Fig 5a and b), the reductions in industrial gas consumption cannot be explained by the
increases in total electricity generation, suggesting that decreased industrial production were likely caused by gas
consumption reductions. Consequently, our results show that across thewas7&UK, 5.7±9.3 TWh of the total industrial
gas reduction (76.3±26.9%) might translated into a lower industrial production (Fig 5c).
**5.Discussion**
**5.1.LNG is a structural alternative to Russian gas**
During the post-invasion winters, the most notable development was the significant increase in LNG imports into the
EU27&UK, which surged from 20.7 % to 37.5% of the total gas supply. EU countries are actively expanding their
LNG import capacities by an additional 13% of the current capacity in the near term (Table S3), positioning LNG as
a structural alternative to Russian pipeline gas. However, the reliability of global LNG supply to EU27&UK remains
uncertain. In 2022, the major global LNG supply increment came from the U.S., accounting for 142 TWh (Global
liquefied natural gas trade volumes set a new record in 2022, 2023) (Australia exports record LNG in 2022:
EnergyQuest, 2023), which was still insufficient to meet Europe's increased winter LNG demands. Our previous
forecasts (Fig 6, right panel) underestimated the importance of additional LNG supply in mitigating the gas crisis
because we assumed constant global LNG demand. Europe therefore may have benefited from substantial reductions
in Chinese LNG demand in 2022 (202 TWh) due to zero COVID-19 measures and renewed energy production from
coal  (How the European Union can avoid natural gas shortages in 2023, 2023) (China's natural gas consumption and
LNG imports declined in 2022, amid zero-COVID policies, 2023) (China fuels economic recovery with higher coal
imports, not LNG, in Q1, 2023). In addition, despite the geopolitical tensions, a substantial volume of LNG imports
continued from Russia (Pécout, 2023).
Continuing and long-term reliance on LNG imports may pose considerable economic and climate risks for Europe.
The cost of gas supplied via LNG is notably higher when compared to pipeline gas, primarily due to the increased
expenses associated with transportation, liquefaction, and regasification (LNG vs. Pipeline Economics [Gaille Energy
Blog Issue 66], 2023). These costs can translate into elevated end-user gas prices, as reflected in the doubling of
household gas and energy costs (Fig S6). Furthermore, LNG tankers have higher transportation-related emissions
compared to pipelines. Our estimations suggest that LNG tankers might produce 2.4 times the amount of $CO_2$
equivalent emission when transporting the same amount of gas via pipeline when considering a potential leakage from
pipeline transportation at 1.4% (see supplementary) (Lelieveld et al., 2005).
Dutch TTF (Title Transfer Facility) natural gas prices (Fig S5) exhibited a sharp rise in the winter of 2022-2023 in
response to the profound supply pattern changes following the Russian invasion of Ukraine, peaked at three times
compared to the pre-invasion levels. Nevertheless, the TTF price has since returned to pre-invasion levels for the



second post-invasion winter (winter of 2023-2024), suggesting a potential alleviation of the gas crisis through existing
gas supply-consumption dynamics.
**5.2.Norway and Northern Africa are stabilizers of gas supply**
The increase in gas supply from Norway and North Africa to Europe during the post-invasion winters was much
smaller than the LNG increment, contributing only 28.9% of the total LNG increase. However, their contributions
were crucial as "stabilizers" in balancing the gas supply shortages within the EU27&UK by redirecting their exports
to those countries that experienced larger reductions in Russian supply and had infrastructural constraints in accessing
extra LNG supply. For instance, Norway redirected its gas exports from France and the Netherlands to Germany,
while North African exporters redirected their gas exports from Spain to Italy, as shown in Fig 3. Notably, in Germany,
the share of Norwegian gas in the total supply jumped from 33.5% to 60.4%, effectively replacing Russian gas as the
primary source. Similarly, Italy increased its reliance on North African gas from 22.7% to 44.1% of its total supply.
**5.3.Germany has reshaped the intra-EU gas transmission**
To address the gas shortage in Germany, significant adjustments of the intra-EU gas transmission network were
observed, involving a net reversal of the historical East-to-West flow direction (Fig 3). Substantial increments of intra-
country transmissions were seen from countries that are equipped with large, preexisting terminals, e.g. Belgium and
the Netherlands, to Germany (Fig 3). New transmission pathways developed to service Germany, indicating close
cooperation among European member states, as shown in Fig 3: 1) transmissions from France to Germany came online
in October 2022 (210 TWh) (Zhou et al., 2023) by resolving the different gas odorization systems between the two
countries, 2) Denmark and Sweden reduced their reliance on Germany transmission by directly importing gas from
Norway. and 3) Germany began its own direct LNG imports in December 2022 (14 TWh) via the newly developed
LNG terminals (Ukraine war pushes Germany to build LNG terminals, 2023). While the changes in the gas
transmission network may not be permanent—pending the establishment of German LNG terminals or a reduction in
its structural dependency on gas—the overall shift in intra-EU gas transmission from Eastern to Western Europe,
previously dominated by Russian supplies, has been structurally reversed.
**5.4.Behavioral heating reduction is economic-sensitive**
In residential heating, consumption reductions were primarily driven by behavioral changes as previously discussed.
The declines in isothermic gas heating consumption during the post-invasion winters (flatter slopes of TGC in Fig S9)
can be associated with the concurrent, rapid surge in heat pump sales within the EU and their increasing adoption in
household heating  (Heat Pumps in Europe – Key Facts and Figures, 2023). The heat pumps, despite their superior
heating efficiency, may not necessarily lead to lower overall residential heating costs. This is partly due to higher
electricity prices caused by increased demand from both residential and industrial sectors using heat pumps (Fig S6
a), which can offset the cost benefits of transitioning from gas heaters. Additionally, lower start heating temperatures
(Fig S9) were observed, indicating the reduced comfort levels for heating inception and were likely driven by rising



gas and energy prices (Fig S6). Therefore, further shifts away from gas to heat pumps in residential heating are
economically sensitive and depend on the dynamics between gas and electricity supply and pricing.

**5.5. Structural independent from gas-powered electricity**

The substantial growth of renewables was found in the post-invasion winters and it dominated the substitution of gas-
powered electricity (Fig 4b). In the initial post-invasion period, fossil fuel substitution remained significant,
particularly in Germany and Italy (Fig S14 a), accounting for 48.1% of the substitution in these two countries. However,
by the second post-invasion winter (winter 2023-2024), renewable energy took the lead across all EU27&UK countries,
with renewables accounting for an increased substitution rate of 114.2%, suggesting the structural shifting from gas-
powered electricity has been successfully developing in EU27&UK.
On the other hand, the contribution of nuclear energy to this substitution was considerably lower than expected due to
the maintenance of the French nuclear reactor fleet and Germany's phase-out of nuclear power (Fig S14). Nevertheless,
nuclear energy might regain strategic importance in offsetting gas-fueled power generation in the future, particularly
as some French reactors come back online and the demand for electricity increases due to widespread heat pump
installations.
The expansion of electricity production through alternative energy sources, such as green electricity and nuclear power,
offers a dual benefit. It not only substitutes for gas-fueled power generation but also supports gas reduction in the
residential heating and industrial sectors through the adoption of heat pumps. Achieving full structural independence
from gas still presents challenges in the near term due to existing electricity infrastructure constraints, transitioning
toward green electricity, and nuclear power maintenance issues.

**5.6. Pathway to EU Energy Security**

Our analysis of the structural and temporary shifts in European gas and energy supply and consumption patterns during
the post-invasion winters underscores the region's institutional and infrastructural resilience to this energy crisis. We
found that energy security has been and will continue to be enhanced by: 1) increasing LNG imports to diversify gas
supply sources, 2) strengthening both international and intra-EU cooperation, and 3) systematically reducing gas
dependency by decreasing residential gas heating and expanding the use of renewable and nuclear power.
While addressing existing challenges on the pathway to the EU energy security, such as the dynamics of the global
LNG market, EU gas infrastructure capacities, and the potential impact on climate change, we also identify key areas
that require further attention (Four challenges of the energy crisis for the EU's strategic autonomy, 2023): 1)
Systematic substitution of heat fuel from gas to electricity will require systematic increases in power supply and
generation capacity; 2) In turn, rising electrical demand will need to be met with expansions in power generation,
preferably through renewable energy sources; 3) Effective energy redistribution, including both gas and electricity,
among EU countries will call for a unified strategy that includes greater integration and enhancement of both physical
and institutional infrastructures.



## 6. Data availability

We updated one dataset (EUGasSC) and published two new datasets (EUGasNet and EUGasImpact) as CSV files, and they are hosted on the Zenodo platform: https://doi.org/10.5281/zenodo.11175364 (Zhou et al., 2024). The datasets are open-access and are licensed under a Creative Commons Attribution 4.0 International license. The column headings of the data dictionary files as well as the unit of each variable are listed in Table S4.

Our datasets provide daily updates on gas supply, storage, transmission, and consumption, providing sectoral and country-specific data on the European gas landscape. Our datasets also capture the pattern changes after the Russian invasion of Ukraine, as well as the driving factors of those changes. These datasets can serve as either input or reference datasets for further research across various fields, including gas/energy modeling, carbon emission studies, climate change impacts, geopolitical policy discussions, and international gas/energy market analysis. By offering multidimensional insights, our data facilitate a comprehensive understanding of the dynamics within the EU gas landscape and contribute to outlining pathways toward EU energy security. Chuanlong Zhou, who collected the data and performed the analysis, and Philippe Ciais, who is an expert on the background of this study, are at the disposal of researchers wishing to reuse the datasets.

## 7. Conclusions

We updated one dataset (EUGasSC) and introduced two new datasets (EUGasNet and EUGasImpact) for the EU27 and UK at daily resolutions: (1) the EUGasSC dataset, describing the sectoral and country-based daily natural gas supply, storage, and consumption, (2) the EUGasNet dataset, describing the intra-EU gas transmission with specified supply source data, and 3) the EUGasImpact, describing the sector-specific gas consumption changes between the pre- and post- invasion winters, combining with the contributions of driving factors. Together, these datasets offer multidimensional insights into the dynamics within the EU gas landscape, and can be valuable to future research on various fields and topics, such as energy modeling, carbon emission analysis, climate change research, and policy discussions.

Using these datasets, we analyzed the pattern changes of the EU gas landscape between the pre- and post-invasion winters. We show how the EU27&UK adapted their gas supply, transmission, and consumption patterns in response to the gas crisis. We quantified the contribution of driving factors in the residential heating, power, and industrial sectors. Our findings indicate significant changes and growing structural independence from gas during the post-invasion winters: 1) total gas consumption decreased by 19.0% due to the sudden loss of Russian gas, 2) LNG emerged as the largest gas supply source, accounting for 37.5% of the total gas supply, 3) Intra-EU gas transmission adjustments focused primarily on addressing the significant shortfall in Germany, 4) behavioral changes in household heating individually contributed most to consumption reduction (28.5%), 5) renewable electricity dominated the substituted gas-powered electricity (92.7%), and 6) relatively large consumption reductions can be associated with declines in industrial production and power shortage (33.9%).

**Figures**

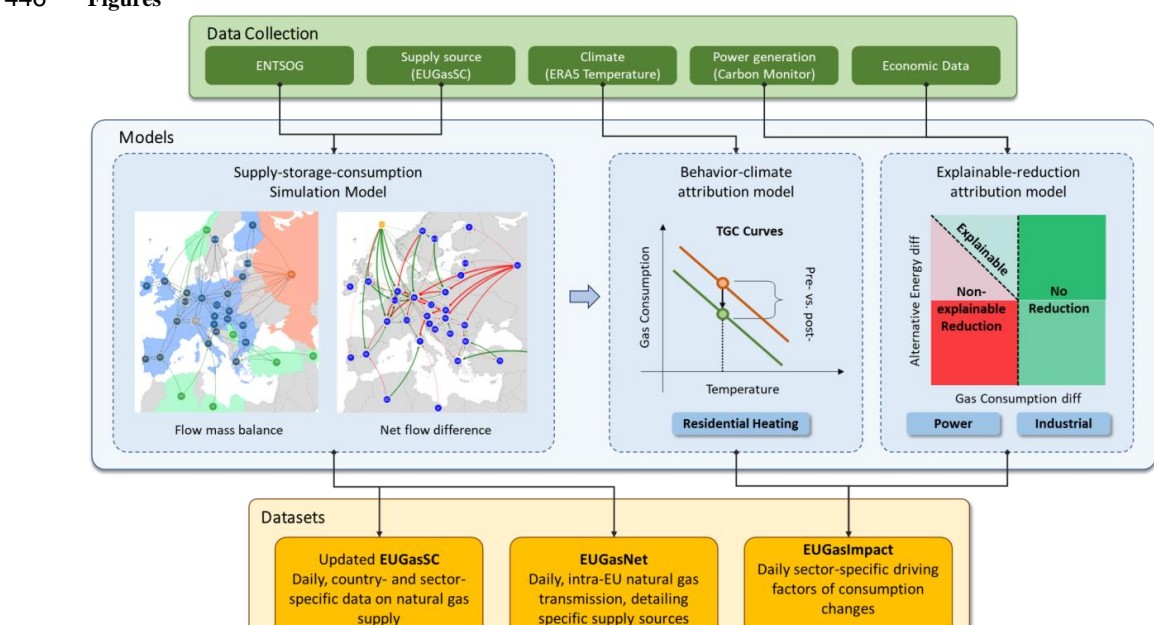

**Fig 1. Study workflow and conceptual framework of this study.** The workflow of this study includes input datasets,
supply-storage-consumption model, sector-specific consumption change attribution models, and three daily output
datasets for the EU gas landscape on supply, transmission, and impacts. The supply-storage-consumption model has
been described in our previous work (Zhou et al. 2023).

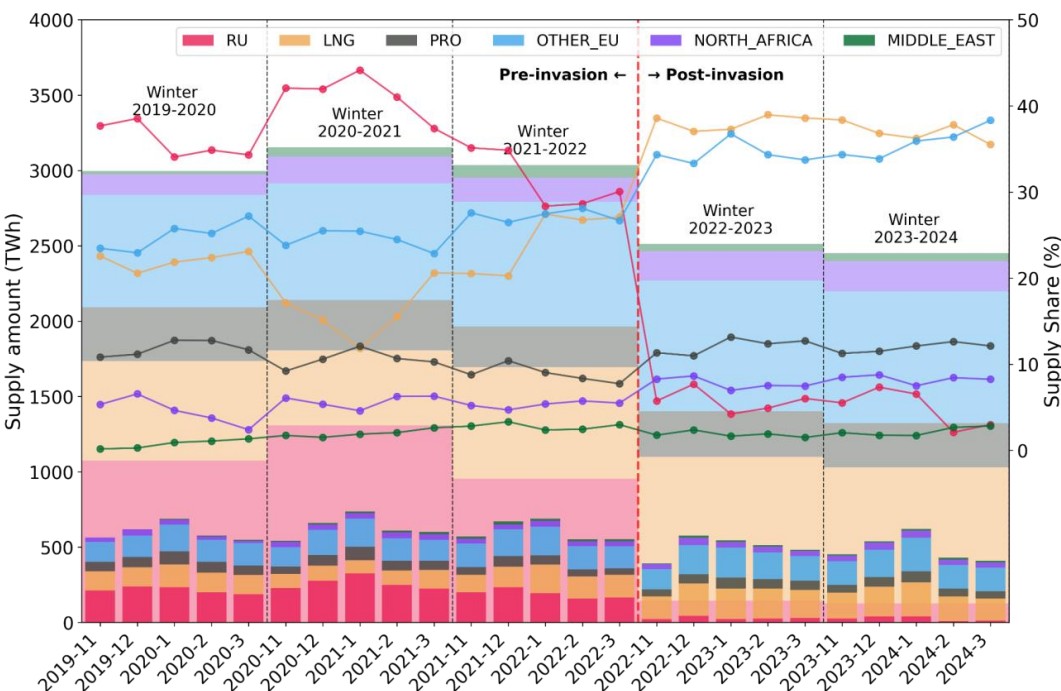

**Fig 2. Winter gas supply in EU27&UK from 2019 to 2024.** This figure displays monthly and winter-aggregated gas
supply amounts (narrow and wide bars, respectively) and their shares (line graphs) from 2019 to 2024. The gas supply
sources are differentiated by color, including pipeline imports from Russia (RU), Norway+Serbia (OTHER_EU),
Libya+Algeria (NORTH_AFRICA), and Turkey+Azerbaijan(MIDDLE_EAST), LNG imports (LNG), and EU local
productions (PRO).

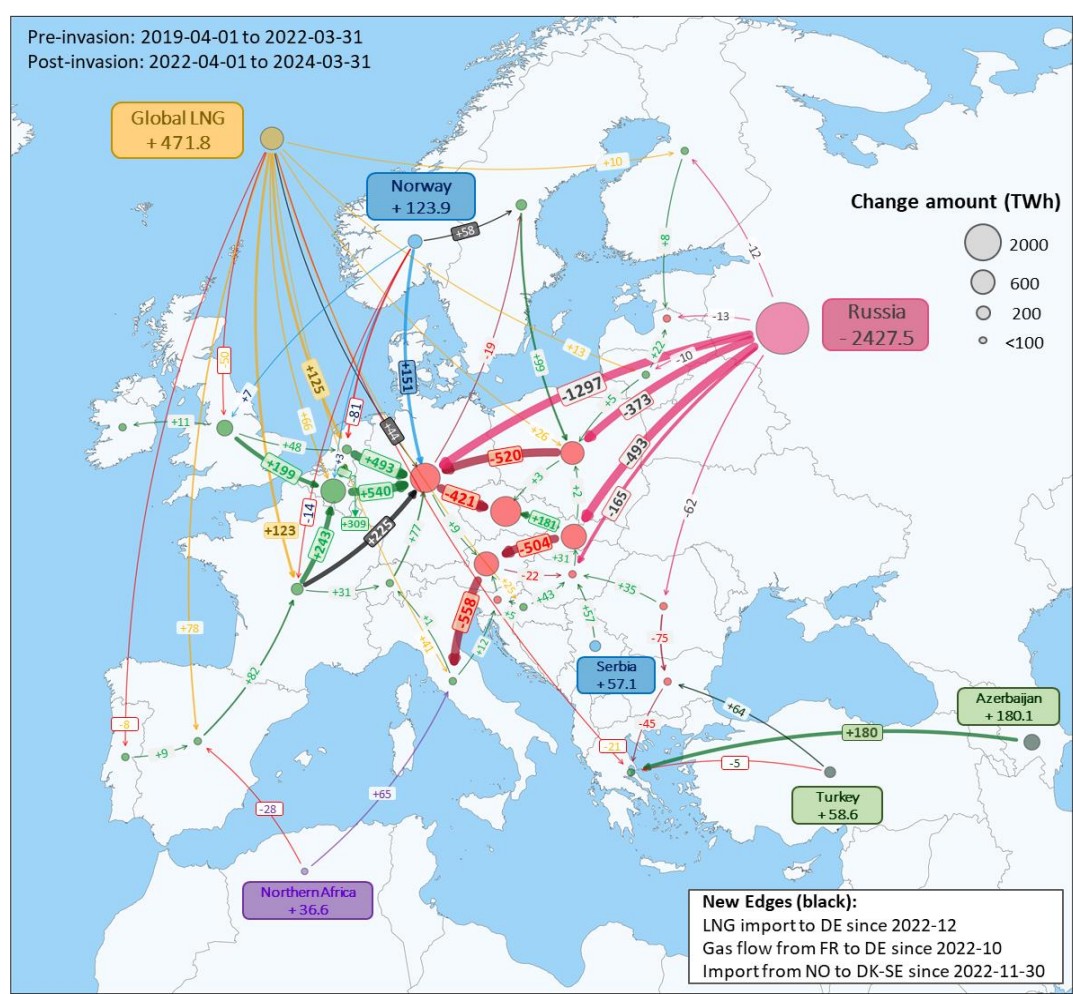

**Fig 3. The annual net flow changes in gas imports and intra-EU transmission network between pre-invasion (2019-04-01 to 2022-03-31) and post-invasion periods (2022-04-01 to 2024-03-31).** The figure illustrates changes by subtracting annual average pre-invasion values from post-invasion values. Positive values indicate increments in the post-invasion period. Circle sizes represent changes in transmission amounts by country (green for increases, red for decreases). Intra-EU transmissions (edges) are color-coded (green for increase, red for decrease) with imports differentiated by import sources. Note that the flow directions are shown based on the major flow directions in the post-invasion period (detail see method 2.4).

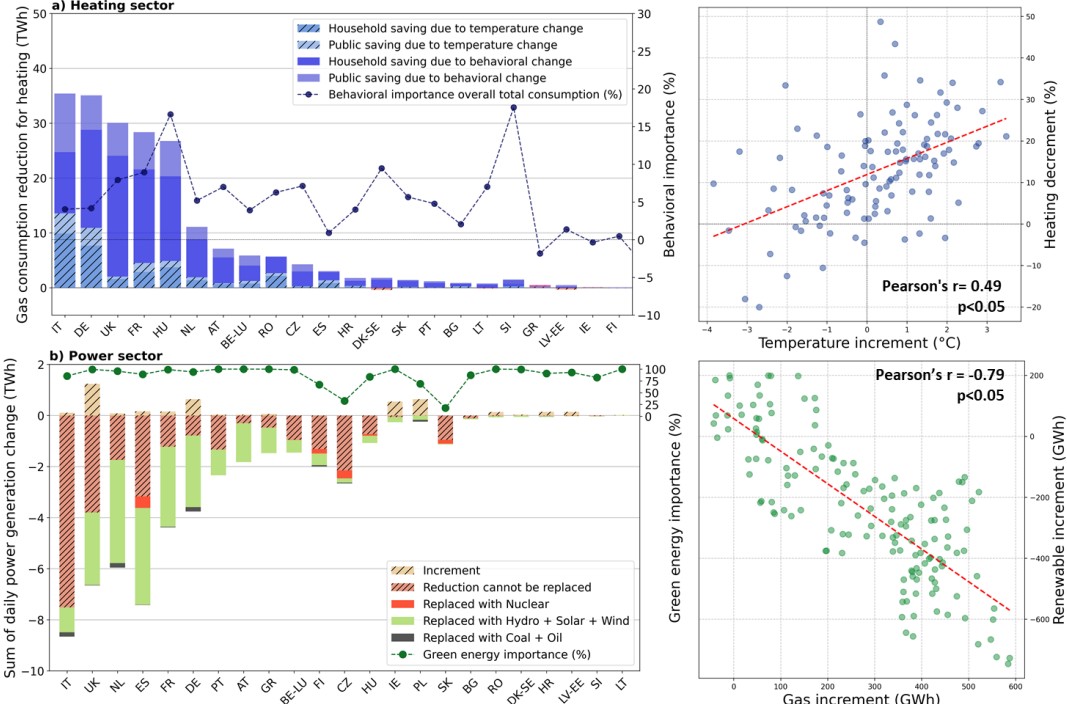

**Fig 4. Consumption change attributions in the residential heating (top) and power sector (bottom) in EU27&UK.** The gas consumption chanes in the heating sector is separated into behavioral change (solid color bars) and climatic change (hatched bars). The daily gas-powered electricity generation change is separated into the replaceable reduction (solid color bars), the non-replaceable reduction (red hatched bars), and the increment (yellow hatched bars). The correlation between temperature increment and heating consumption reduction is shown in the top-right panel. The correlation between gas-powered electricity increment and renewable power increment is shown in the bottom-right panel.



Earth System
Science
Data

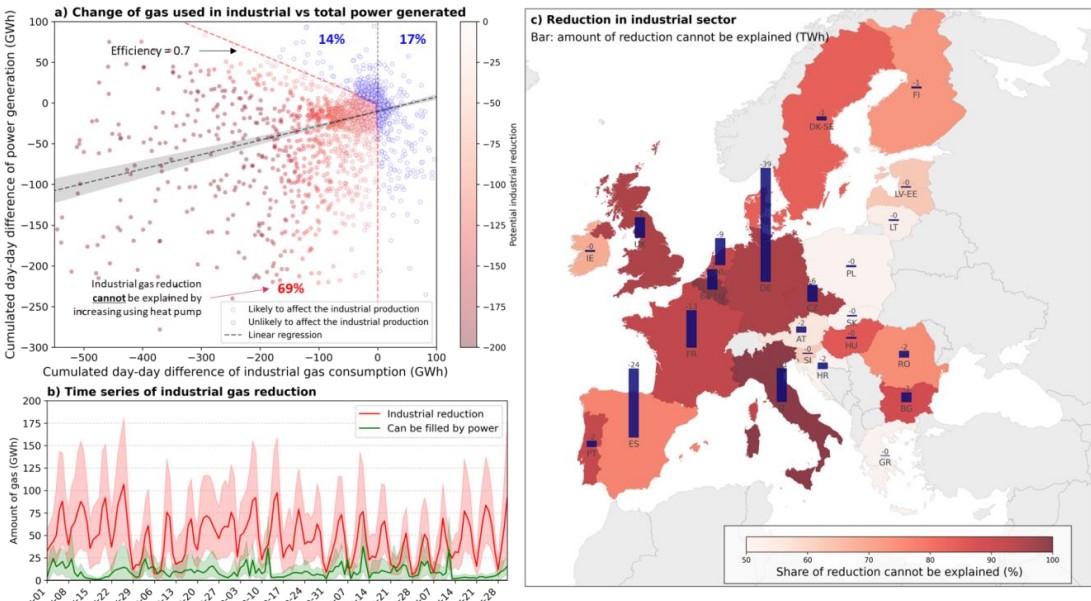

**Fig 5. Analysis of potential impact on industrial productions**. In the subplots: a) explainable-reduction attribution model with daily comparisons of gas consumption in industrial and total electricity generation, b) time series of industrial gas reduction and reduction that can be filled with electricity generation, and c) the amount (bar) and proportion (choropleth) of reductions that potentially impact industrial production. R2 of the linear regression is 0.38 with a p-value < 0.01.

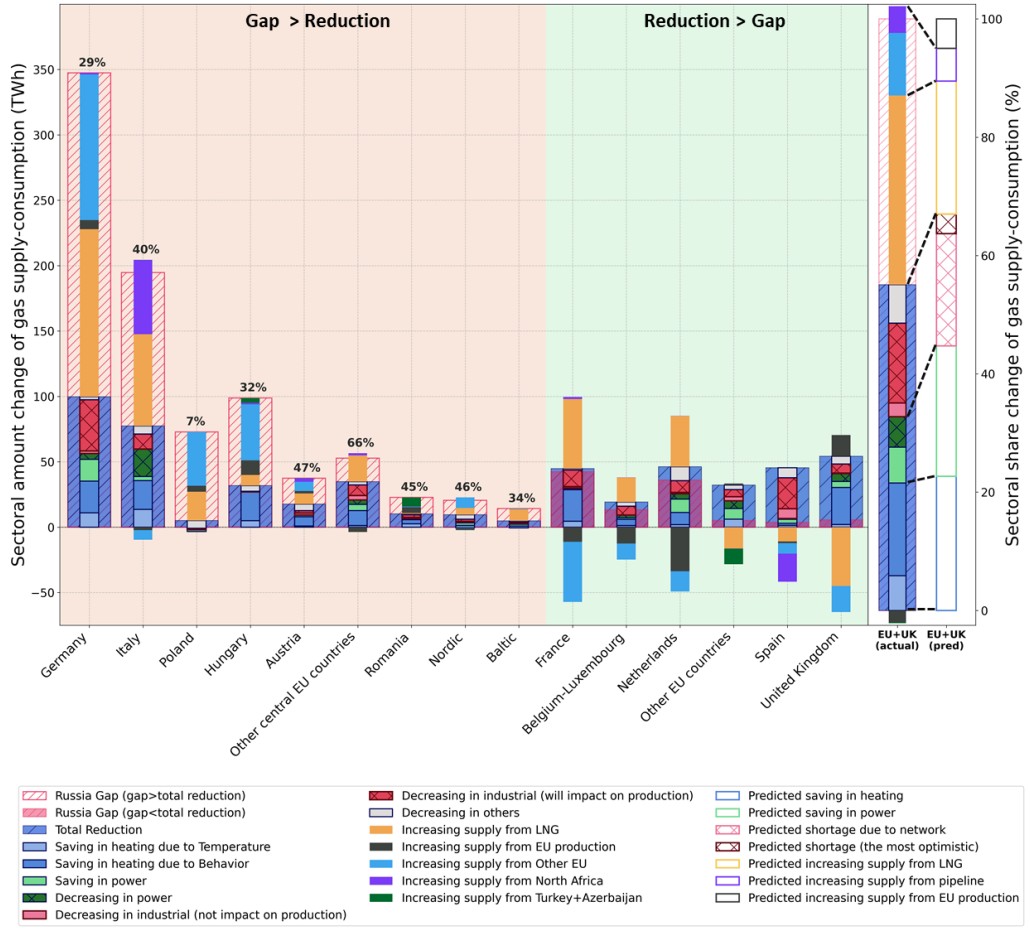

**Fig 6. The gas supply-consumption pattern changes compared between post-invasion and pre-invasion winters in the EU27&UK.** The differences are calculated by subtracting the winter average of pre-invasion winters from post-invasion winters. Wide bars (left panel) represent total gas consumption reduction (blue hatching) and Russian supply reduction (red hatching). Narrow bars illustrate supply (solid color, no borders) and consumption (solid color, borders) attributions of these reductions. Bars with crosses indicate consumption reductions potentially causing negative impacts in the power or industrial sectors. The top numbers denote the percentage of the gap absorbed within the region. The left panel shows the aggregated values and comparisons between our previous estimations. "Baltic" includes Estonia, Latvia, and Lithuania. "Nordic" includes Denmark, Sweden, and Finland. "Other central EU countries" include Slovakia, Slovenia, Czechia, and Croatia. "Other EU countries" include Ireland, Bulgaria, and Portugal.



**Competing interests**

The contact author has declared that none of the authors has any competing interests.

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
