# Peer review of "transmission-consumption structures and driving factors from 2022 to 2024"

_Earth System Science Data, 2024_

## Referee Comment (RC1)

Comments on Europe gas data

Overall, a very good product with easy-to-use data. Needs some 'fixing' before publication in ESSD/

Paragraph beginning at line 68: here authors report attempts to differentiate - for heating, power and industrial sectors - whether reduced energy consumption by sector derives from behavioral, meteorological, source or efficiency changes. They use, variously, behavior-climate attribution models (driven with ECMWF land temperature data) for heating, gas consumption with CM electricity data (for power), etc. For industrial sector they assume incorporation of heat pumps does not decrease industrial production but conclude with statement that "industrial consumption reduction could be explained by the increments of total electricity generation with certain conversion efficiency". Not clear what this statement refers to? This particular approach "similar" to approaches applied for heating or power sectors? Slight revision could clarify? Does this assumption impact subsequent discussions of uncertainties and biases? Same topics discussed with more detail in paragraph starting at line 174. Assumption clarified (somewhat) but still without attribution? Later paragraph refers to specific (= constant over this time period?) gas-to-electricity conversion ratio. Based on Fig S16, which unfortunately does not quantify this relationship, authors assumed a conversion ration of 1:1? Authors allow this reader to stumble over this issue; needs clarification?

Line 83: "LNG imports replaced Russian pipeline imports and become the primary gas supply source to the EU." But, in abstract and throughout paper and data sets reader learns that LNG represented 38% (37.5) of total gas supply. Largest single source, perhaps, but not the "primary" source?

Line 108: Periods of Analysis. Good. But, EUGasSC and EUGasNet files both start with data from 2016? Not explained here nor mentioned elsewhere in this manuscript nor in associated supplement?

Line 188, "0.55±0.21": Readers need to know what '+0.21' means? SD? 95CI? Authors should use identical designation throughout, defined clearly at first use.

Line 188: Fig S9 includes two panels, not one. No composite r2 available? This reader doubts composite as high as 0.55 and gets nothing from 9a or 9b that conveys "effective capture" of TGC data?

Line 191: Reference here "previous study" means Zhou et al 2023? Use citation? So 'overage' in Fig 6, right panel, represents - at top - <5% overproduction? But, authors apparently ignore equal-sized overage at bottom of that panel? Something wrong somewhere?

Line 195 and remainder of paragraph: if pre- vs post- energy uses remained constant, not clear to this reader why that constancy would necessarily lead to overestimation of shortages or drops in production? Need clarification here? Increased energy prices or conservation would lead to same overestimation? Reader assumes grid "interconnection" refers to connections across EU countries but sentence eventually refers to "supply and demand" changes within country? Again, clarification needed?

Lines 199 to 201: Nice list but given these assumptions and residual uncertainties, plus lack of proper validation (probably impossible for these particular data - authors should comment?), this reader feels uncomfortable with numbers to 3 sig figs, e.g. 36.3; authors should round to two sig figs max, throughout?

Paragraph covering lines 206 to 214: Some numbers make sense (add correctly) here but authors would make such checks much easier by consistently adopting either "per winter" or "two post-invasion winters". Mixing one vs two years adds confusion?

Line 215, "decrease in gas consumption (25.5±16.0%)": One year, two year, winter, annual?

Line 228: These represent annual numbers (-2427.5) compared to earlier (976.8, line 206) numbers that represented winter-only? Please ensure that readers know when authors report annual vs winter-only data? Also not clear when authors refer to Western EU countries (France?, Spain?) vs Eastern EU countries (Hungary?, Poland?). Where do Germany, Austria, Italy fit in geographic dichotomy? This reader finds (line 225) delineation of "Central" as well? Nowhere defined? Need a table or map?

Line 315, authors reference gas-electricity conversion efficiency of 0.7. As in Fig 5 but not in Fig S16?

Line 318 - typos

Line 364, "resolving the different gas odorization systems": did this significant accommodation occur only for gas transported eastward from France to Germany, or for all French gas usage? In prior manuscript (Zhou et al 2023) odorization policies represented a significant barrier? Did author's predictions anticipate this change? Significant but not largest among adjustments in Germany but highly significant as a French adjustment?

Line 381: interesting conclusion! Do authors need to apply some uncertainly here? Or, strong prediction?

Line 407, citation lists "Four" but authors list only 3. Readers might prefer to trust authors but will need to understand basis for apparent change?

Line 413, Data availability: Data very good, very easy to find, download and use. Good documentation! Compliments to authors.

Line 453, Figure 2: colors applied in wide bars too dilute? E.g. particularly the gray/black 'Pro' bars, diffuse in wide bars. From constancy reader can assume 'Pro' but not clear enough to un-careful readers?

Line 459, Figure 3: data availability would certainly support reader reproduction of this figure. I started in QGIS, did not finish to this quality but can see how authors came up with these numbers and this figure. Authors should assure readers that open-access tools (e.g. QGIS) will work, even if they used e.g. ArcGIS?

Line 467, Figure 4: "Green energy importance %" not defined nor quantified. Reader must assume that persistent values near 100% represent max possible utilization?

Line 471, referring to Figure 4: not 'yellow' in the version I see?

Line 475, Figure 5: Readers needs to know what color bands around lines in panel b mean? Range, SD, what? Emphasize scale in panel C? No EU country can explain more than 50% of reduction. Major ('Central'?) EU countries - Germany, France, UK, Italy, Belgium - must admit 80% of reductions not explainable? Uncertainty conveyed here not repeated in narrative?

Line 481, Figure 6: Interesting, perhaps key, figure. Takes too long to understand. Very difficult to reproduce. Black block at bottom of right panel already questioned (see above, ref. Line 191). Need some indication of uncertainties? E.g. Baltic or BeneLux: can authors expect readers to rely on such small numbers? Gray or blurred zone must exist between Gap > reduction countries and reduction > gap countries? Authors could assist readers with small attention to details of this figure?

References should - where possible - point to original DOI? Journal will know proper format for references, e.g. when "n.d." might prove appropriate and for numerous EU organizational or news media citations (not trackable nor available to this reader unless they carry DOI).

Confusion in references at end: Zhou et al 2024 (Zenodo data) vs Zhou et al 2023 (prior ESSD paper) not in correct order?

Supplement:

Fig S1: Bars in this graph need to show errors/uncertainties? Looking at bottom (3rd) panel, this reader estimates that only Germany, Hungary, France, Netherlands and UK engaged in significant storage? In every other EU country, data remain within noise limits?

Fig S2: Strong need for uncertainties! For this reader on quick glance: a) no sig differences any sector during summer; and b) authors have not shown, here at least, that pre-war differs statistically from post-war?

Fig S3: Interesting chart. This reader sees France with neither strong need nor strong source? How then did France come up with 225 TWh (Figure 3) for export to Germany?

Fig S4: Date units in top panel not correct? In lower panel, readers learn that storage accumulates during EU summers but supply for that storage arrives during winter? Evidence for annual "flow mass balance" but not for seasonal or monthly? Need to see uncertainties around these graphs?

Table S2: In this data, France shows significant (?) increase in transmission over within-country consumption. Most French export went to Germany? Because authors present data as ratios, readers can't, from this Table at least, determine amounts?

Fig S5: From this graph, reader concludes that Nordstream disruptions had larger impact, in terms of price, than invasion? Time required for sanctions / interuptions to impact prices? Additional source(s) of uncertainty?

Fig S6: Reader needs information about uncertainty bands around gas and electricity prices?

Fig S7: Arrow widths provide only weak indicators? This figure does not show significant export France to Germany?

Fig S8: Daily flows clearly show impact of invasion but, for some exchanges, with lags of up to 6 months? One can get some info about temporal uncertainties from this figure?

Figs S9a, S9b: need author declaration of pre-(orange?) and post-(blue?).
Why does Portugal data show two separate clusters, unlike (for example) Spain?

Fig S11: No uncertainties here? From panels shown in S10, reader has no confidence in values below 0.3, or 0.5?

Fig S12: Again, no uncertainties? This reader concludes that e.g. Austria used high proportions of renewables prior to invasion and that those proportions did not change post-invasion?

Fig S13: Reader needs information about uncertainties represented by color bands?

Fig S14: Difficult but important figure? For this reader, only Germany, Italy, France and UK, perhaps with Spain and perhaps with Austria, showed significant changes over two winters? Everything else (all other countries) within noise?

Fig S15: Uncertainties? Reader finds this figures potentially useful but not without some indication of uncertainty?

Table S4: Very useful! Should appear as Appendix, referenced on Zenodo and in section 6?

---

## Author Comment (AC1)

**RC1**

**General Response**

We are grateful for the reviewer's acknowledgment of the paper's contribution to understanding the shifts in Europe's gas landscape. We particularly thank you for your thorough review and detailed comments. Those comments are greatly helpful to enhance the clarity and impact of our work.

The reviewer's comment focus on two major aspects of the manuscript, mentioned multiple times throughout the comments, that require clarification and improvement:
1. **Clarification of assumptions made in the paper**
2. **Improvement of uncertainty analysis, reporting, and visualization**

Before addressing individual comments, we would like to provide an overview of major assumption we made for sectoral analysis as well as the uncertainty and limitations. We made assumptions and developed models for the residential, power, and industrial sectors, respectively. Those models aim to analyze the driving factors behind consumption changes and their consequential impacts.

**Residential Heating Sector**

To analyze gas consumption changes in the residential heating sector, we employed temperature-gas-consumption (TGC) curves to distinguish between reductions caused by temperature variations and those driven by behavioral changes. This approach aligns well with observed data, as indicated by strong correlation coefficients ($r^2$ values) and the close agreement between predictions from our previous study and real-world data in this analysis. However, there are limitations in our current approach. This model does not fully capture reductions in "real indoor temperatures" due to high energy costs, which may have led to lower heating comfort levels for consumers. Also, the transition from gas-based heating to heat pumps is not explicitly modeled, although we discuss this shift qualitatively in the paper. We will clarify these limitations in the revised manuscript and discuss potential implications for long-term household energy consumption patterns.

**Power Sector**

For the power sector, we combined a separate power generation dataset, CM Power, which provides hourly-resolution data on electricity generated from different energy sources. This dataset was used to analyze potential electricity deficits caused by the gas crisis. In our approach, we assumed that overall electricity demand remained consistent between the pre- and post-invasion periods. To smooth short-term variations, we aggregated the data into 7-day periods for analysis. For each period, we first identified whether gas-powered electricity generation had decreased. If so, we assessed whether surplus electricity from alternative sources was available during the same period to offset the decrease. Any shortfall not covered by surplus electricity was categorized as a "deficit" caused by the gas crisis.

By assuming constant power demands, our estimate of electricity deficits may be overestimated, as the total power generation decreased during the post-invasion period, potentially implying reduced electricity demand. However, it is also interesting to note that these "reduced demands" could reflect the broader impacts of the gas crisis on the EU's living and economic conditions. Our aim is to offer a simplified analysis to quantify potential deficits under the assumption that EU countries sought to maintain pre-invasion living standards or economic activity levels.

**Industrial Sector**

For the industrial sector, we aimed to differentiate gas consumption changes caused by energy use from those caused by raw material use, assuming that reductions in material use would negatively impact industrial production. Due to limited data availability, we made stronger assumptions for this sector: 1) Industrial production levels were assumed to remain constant (implying that raw material consumption would also remain constant), 2) All reductions in gas consumption were assumed to come from energy use, primarily for industrial heating, 3) To maintain industrial production, heating energy reductions would need to be replaced by electricity.

We used a similar method to the power sector to analyze whether surplus electricity could offset the reduction in industrial gas consumption with a gas-to-electricity conversion factor.

The major limitation of this approach is that we do not include other fuel types, such as oil or coal, that may have been used for industrial energy. This will make our result overestimate the impact. Additional, the surplus electricity would be entirely available for industrial use, which may not always hold true.

Although our approach may overestimate the actual impact on industrial production, this approach explores a worst-case scenario—assuming no other fossil fuels were available as alternatives—and estimates how much industrial gas consumption reduction might lead to decreased production. The findings can also highlight the additional electricity demand required for a transition to greener energy sources in the industrial sector. Electrifying industrial heating, particularly through heat pumps, would necessitate a substantial expansion of clean electricity supply. This underscores the critical need to scale up renewable energy generation and strengthen grid capacity to ensure that industrial production remains stable while reducing reliance on fossil fuels.

These assumptions and methodologies provide a clear framework for analyzing sector-specific consumption changes and their penitential drivers and impacts.

**Detailed comments**

**Paragraph beginning at line 68: here authors report attempts to differentiate - for heating, power and industrial sectors - whether reduced energy consumption by sector derives from behavioral, meteorological, source or efficiency changes. They use, variously, behavior-climate attribution models (driven with ECMWF land temperature data) for heating, gas consumption with CM electricity data (for power), etc.**
Addressed in the general response.

**For industrial sector they assume incorporation of heat pumps does not decrease industrial production but conclude with statement that "industrial consumption reduction could be explained by the increments of total electricity generation with certain conversion efficiency". Not clear what this statement refers to? This particular approach "similar" to approaches applied for heating or power sectors? Slight revision could clarify?**
As outlined in our general response, our analysis of the industrial sector aimed to distinguish reductions in gas consumption for energy use (industrial heating) from reductions due to raw material use. To achieve this, we applied an explainable-reduction attribution model with day-to-day change comparisons, similar to the approach used in the power sector.

The key difference between the two sectors: 1) In the power sector, we compared electricity generation from gas-fired plants with electricity generation from alternative sources (i.e., electricity-to-electricity comparison), 2) In the industrial sector, we examined whether increased

electricity generation could compensate for the reduction in industrial gas consumption for heating (i.e., electricity-to-gas comparison). Since these two forms of energy are not directly comparable, we introduced a gas-to-electricity conversion efficiency to estimate the extent to which electricity could replace industrial gas consumption.

To further clarify this point, we have revised the text in the Introduction as follows:

*"Similar day-to-day change comparison approach is performed to assess whether the industrial consumption reduction could be explained by the increments of total electricity generation with a gas-to-electricity conversion efficiency."*

In the method section, 2.5.3 Industrial sector, add the following text to clarify the usage of conversion efficiency:

*"In the industrial sector, we examined whether the increase in electricity generation could offset the reduction in industrial gas consumption for heating (electricity-to-gas comparison). Since electricity and gas are not directly interchangeable, we applied a gas-to-electricity conversion efficiency to estimate the potential replacement effect."*

**Does this assumption impact subsequent discussions of uncertainties and biases?**

The uncertainties and limitations associated with these assumptions are addressed in our general response. To further clarify their implications, we have modified the following text to Section 3: Uncertainties and Bias:

*"In the industrial sector, our simplified assumption does not account for the substitution of gas with other fossil fuels as energy source, such as oil or coal, due to the lack of reliable data, even though these fuels were widely used by industries during the energy crisis to avoid disruptions. Additionally, many industrial processes require high temperatures that heat pumps alone cannot provide. As a result, our analysis likely overestimates the impact of the gas crisis on industrial production. However, it serves as a worst-case scenario, providing an upper-bound estimate of potential industrial production losses without considering alternative fossil fuels (e.g., oil or coal) and relying solely on electricity, which implies the additional electricity demand required for industrial electrification in the transition to greener energy sources."*

**Same topics discussed with more detail in paragraph starting at line 174. Assumption clarified (somewhat) but still without attribution? Later paragraph refers to specific (= constant over this time period?) gas-to-electricity conversion ratio. Based on Fig S16, which unfortunately does not quantify this relationship, authors assumed a conversion ration of 1:1? Authors allow this reader to stumble over this issue; needs clarification?**
The assumption and attribution are addressed in both our general response and detailed response above. To clarify, we did not assume a 1:1 gas-to-electricity conversion ratio. Instead, we used a conversion ratio of 0.7, as shown in Figure 5a.

Regarding Figure S16, it is intended as a conceptual illustration rather than a quantitative representation of the conversion ratio. To prevent confusion, we add "*(concept illustrated in Fig S16)*" instead of direct refer to Fig S16.

**Line 83: "LNG imports replaced Russian pipeline imports and become the primary gas supply source to the EU." But, in abstract and throughout paper and data sets reader learns that LNG represented 38% (37.5) of total gas supply. Largest single source, perhaps, but not the "primary" source?**

Thanks for the suggestion, as LNG can be imported from different countries, so we modify the text as "the largest gas supply source to the EU". And we are currently working on a separate study that tracks LNG tankers to allocate LNG supply sources in detail to EU countries. With this study, we might be able to attribute whether there is a single largest LNG source to EU.

**Line 108: Periods of Analysis. Good. But, EUGasSC and EUGasNet files both start with data from 2016? Not explained here nor mentioned elsewhere in this manuscript nor in associated supplement?**

Our EUGasSC and EUGasNet datasets indeed start in 2016, but for our analysis, we focus only on the three pre-invasion winters and two post-invasion winters to assess the impact of the energy crisis. To clarify this, we have added the following text to Section 6: Data Availability:
*"The EUGasSC and EUGasNet datasets are available from 2016, while the EUGasImpact dataset is available only for the two post-invasion winters."*

**Line 188, "0.55±0.21": Readers need to know what '+0.21' means? SD? 95CI? Authors should use identical designation throughout, defined clearly at first use.**

In this paper, all uncertainty values presented using the "±" notation indicate standard deviations (SD). To clarify this, we have added the following sentence at the first occurrence of uncertainty reporting: "All values expressed as '±' in this paper represent standard deviations (SD)."

**Line 188: Fig S9 includes two panels, not one. No composite r2 available? This reader doubts composite as high as 0.55 and gets nothing from 9a or 9b that conveys "effective capture" of TGC data?**

The Fig S9 have two panels, including the household and public building cases in EU countries. We could use composite r2, using gas consumption as the weight for each component, however, this will overestimate the model performance. This is because the countries that consume more gas, such as DE, and FR, have higher r2. We want to show although our model can effective capture the TGC curve, however, it might have lower performance in those counties with less gas consumption (usually warmer countries, such as PT and ES)

**Line 191: Reference here "previous study" means Zhou et al 2023? Use citation? So 'overage' in Fig 6, right panel, represents - at top - <5% overproduction? But, authors apparently ignore equal-sized overage at bottom of that panel? Something wrong somewhere?**

Yes, the "previous study" refers to Zhou et al. 2023, and we will add citation. Regarding Figure 6 (right panel), this discussion focuses specifically on the heating sector and aims to demonstrate that our TGC model effectively predicts gas consumption changes. We compare our pre-invasion predictions (from Zhou et al. 2023) with post-invasion observations to assess model accuracy.The right panel of Figure 6 illustrates how the shortfall in Russian gas supply was addressed across different sectors, compared to our previous projections. The grey bar below zero represents the unexpected decline in EU gas production, which is opposite to our original expectations. The magnitude of this decline is equal to the overage in the wide bar (Russian gas shortfall), ensuring that the supply-demand balance remains consistent. Thus, there is no error in the figure.

**Line 195 and remainder of paragraph: if pre- vs post- energy uses remained constant, not clear to this reader why that constancy would necessarily lead to overestimation of shortages or drops in production? Need clarification here? Increased energy prices or conservation would lead to same overestimation? Reader assumes grid "interconnection" refers to connections across EU countries but sentence eventually refers to "supply and demand" changes within country? Again, clarification needed?**

We appreciate the reviewer's request for clarification. The assumption of constant total power generation and electricity demand was made to simplify the comparison between pre- and post-invasion winters. However, this simplification may overestimate negative impacts (e.g., power shortages or declines in industrial production) because electricity demand likely decreased due to high prices and conservation efforts. Since our model does not account for these demand-side changes, some of the estimated "shortages" may actually reflect lower demand rather than true supply constraints. The discussion on "interconnection" refers to another limitation of our analysis, we did not account for cross-border electricity transmission within the EU, which may have helped mitigate localized electricity supply imbalances. We have revised the text as follows:

*"However, assuming unchanged electricity demand overlooks demand variations driven by rising electricity prices and energy conservation measures across the EU (What measures are European countries taking to conserve energy?, 2023) (Averting Crisis, Europe Learns to Live Without Russian Energy, 2023). This could lead to an overestimation of "negative impacts," as some observed reductions may stem from lower demand rather than actual supply constraints. Another limitation of our approach is that we did not account for cross-border electricity transmission within the EU, which could have played a role in balancing supply and demand at the national level."*

**Lines 199 to 201: Nice list but given these assumptions and residual uncertainties, plus lack of proper validation (probably impossible for these particular data - authors should comment?), this reader feels uncomfortable with numbers to 3 sig figs, e.g. 36.3; authors should round to two sig figs max, throughout?**

We appreciate the reviewer's concern regarding numerical precision, which is indeed very imprtant. This issue was also raised by other reviewers. We have carefully reviewed all numerical values presented in the manuscript and have, and reduced the significant figures to two for model-derived values.

**Paragraph covering lines 206 to 214: Some numbers make sense (add correctly) here but authors would make such checks much easier by consistently adopting either "per winter" or "two post-invasion winters". Mixing one vs two years adds confusion? Line 215, "decrease in gas consumption (25.5±16.0%)": One year, two year, winter, annual?**

In our study, we aim to present the two post-invasion winters together as a single period whenever there is no significant difference between them, as we consider them part of the first post-invasion phase. We have modified the text to consistently refer to the two post-invasion winters together, avoiding ambiguity. For "decrease in gas consumption (25.5±16.0%)", this represents the average percentage reduction across all EU countries over the two post-invasion winters.

**Line 228: These represent annual numbers (-2427.5) compared to earlier (976.8, line 206) numbers that represented winter-only? Please ensure that readers know when authors report annual vs winter-only data? Also not clear when authors refer to Western EU countries (France?, Spain?) vs Eastern EU countries (Hungary?, Poland?). Where do Germany, Austria, Italy fit in geographic dichotomy? This reader finds (line 225) delineation of "Central" as well? Nowhere defined? Need a table or map?**

There are two major concerns:

1. Clarifying Time References

Our study takes an annual perspective when analyzing net intra-EU gas transmission because gas flows in non-heating seasons are also important, which is stated in the 2.2 Periods of analysis. This distinction is illustrated in Figure 3 and clarified in the text to prevent confusion.

2. Removing Ambiguous Regional Labels

We used the terms Western, Eastern, and Central EU to describe gas transmission flow directions, and they are not used for group analysis. Additionally, we provided two critical pathways of reduced net Russian gas transmission:

*Russia → Slovakia → Austria → Italy*

*Russia → Poland → Germany*

To reduce confusion, we have removed these terms and revised the text as follows:

"*During the pre-invasion period, the dominant gas transmission direction was from East-Central Europe toward Western Europe. … In response, gas transmission in the reverse direction increased, compensating for reduced Russian supply—for example, gas flows from Spain-France to Germany (Fig. 3, green lines and circles).*"

**Line 315, authors reference gas-electricity conversion efficiency of 0.7. As in Fig 5 but not in Fig S16? Line 318 - typos**

This issue has been addressed above. The gas-electricity conversion efficiency of 0.7 is explicitly referenced in Figure 5, while Figure S16 serves only as a conceptual illustration rather than a quantitative representation. The typos in Line 318 have been corrected.

**Line 364, "resolving the different gas odorization systems": did this significant accommodation occur only for gas transported eastward from France to Germany, or for all French gas usage? In prior manuscript (Zhou et al 2023) odorization policies represented a significant barrier? Did author's predictions anticipate this change? Significant but not largest among adjustments in Germany but highly significant as a French adjustment?**

The resolution of gas odorization differences specifically applied to gas transported eastward from France to Germany, not to all French gas usage. In our previous study (Zhou et al., 2023), we identified odorization policies as a major technical barrier preventing seamless France-Germany gas flow by comparing both scenarios with and without full transmission capability between the two countries.

Since this transmission capability was only recently established, we still observe a transmission "bypass" through France-Belgium-Germany, as shown in Figure 3. This suggests that while direct transmission has improved, some cross-border gas flows still rely on alternative pathways.

*"The resolution of gas odorization differences enabled direct France to Germany gas transmission, overcoming a longstanding technical barrier (Zhou et al., 2023), though a*

*significant bypass route through France-Belgium to Germany remains visible in Fig. 3, highlighting ongoing efforts toward full network integration."*

**Line 381: interesting conclusion! Do authors need to apply some uncertainly here? Or, strong prediction?**

We appreciate the reviewer's comment. But this statement is not based on a numerical calculation, and we do not quantify uncertainty in this particular conclusion.

**Line 407, citation lists "Four" but authors list only 3. Readers might prefer to trust authors but will need to understand basis for apparent change?**

"*(Four challenges of the energy crisis for the EU's strategic autonomy, 2023)*" is a reference. We conclude three key areas that require further attentions. We will fix the reference format if there are any mistakes.

**Line 413, Data availability: Data very good, very easy to find, download and use. Good documentation! Compliments to authors.**

Thanks!

**Line 453, Figure 2: colors applied in wide bars too dilute? E.g. particularly the gray/black 'Pro' bars, diffuse in wide bars. From constancy reader can assume 'Pro' but not clear enough to un-careful readers?**

We modified the PRO with Production and improve the color of the figure.

**Line 459, Figure 3: data availability would certainly support reader reproduction of this figure. I started in QGIS, did not finish to this quality but can see how authors came up with these numbers and this figure. Authors should assure readers that open-access tools (e.g. QGIS) will work, even if they used e.g. ArcGIS?**

The figure is generated with python code. We can share the code for it along with the dataset.

**Line 467, Figure 4: "Green energy importance %" not defined nor quantified. Reader must assume that persistent values near 100% represent max possible utilization? Line 471, referring to Figure 4: not 'yellow' in the version I see?**

To improve clarity, we defined "Green energy importance %" in the figure caption as:
"*The share of green energy in the substituted gas-powered electricity.*"
Updated the color reference from "yellow" to "beige" to accurately reflect the color shown in the figure.

**Line 475, Figure 5: Readers needs to know what color bands around lines in panel b mean? Range, SD, what? Emphasize scale in panel C? No EU country can explain more than 50% of reduction. Major ('Central'?) EU countries - Germany, France, UK, Italy, Belgium - must admit 80% of reductions not explainable? Uncertainty conveyed here not repeated in narrative?**

As discussed in the general response and detailed response above, our results represent a worst-case scenario, which may overestimate the impact of reductions. Rather than treating these values as direct impact to the industrial production, they should be rather considered as the electricity deficit required to fully replace fossil fuel use (gas, oil, and coal) in the power sector.

The color bands in Figure 5, panel (b) represent the 95% confidence interval (CI) across countries. We have added this clarification to the figure caption to ensure readers can correctly interpret the data.

**Line 481, Figure 6: Interesting, perhaps key, figure. Takes too long to understand. Very difficult to reproduce. Black block at bottom of right panel already questioned (see above, ref. Line 191). Need some indication of uncertainties? E.g. Baltic or BeneLux: can authors expect readers to rely on such small numbers? Gray or blurred zone must exist between Gap > reduction countries and reduction > gap countries? Authors could assist readers with small attention to details of this figure?**

Black block at bottom issue is addressed above.
Uncertainties cannot be added to Figure 6, as the figure is already highly detailed and complex. To improve readability and better assist readers in interpreting the data, we have added a zoom-in inset figure for the small bars (e.g., Baltic, Benelux regions) to ensure their values are clearly visible.

**References should - where possible - point to original DOI? Journal will know proper format for references, e.g. when "n.d." might prove appropriate and for numerous EU organizational or news media citations (not trackable nor available to this reader unless they carry DOI).**
**Confusion in references at end: Zhou et al 2024 (Zenodo data) vs Zhou et al 2023 (prior ESSD paper) not in correct order?**

We will revise the reference format throughout the paper to ensure consistency and compliance with the journal's guidelines.

**Supplement:**
**Fig S1: Bars in this graph need to show errors/uncertainties? Looking at bottom (3rd) panel, this reader estimates that only Germany, Hungary, France, Netherlands and UK engaged in significant storage? In every other EU country, data remain within noise limits?**

The error bars will be added. This figure shows the difference between the Post-invasion and pre-invasion periods, so the panel 3 shows are countries who increase their storage in the post-invasion periods. The value equals 0 does not mean there is no storage in those counties.

**Fig S2: Strong need for uncertainties! For this reader on quick glance: a) no sig differences any sector during summer; and b) authors have not shown, here at least, that pre-war differs statistically from post-war?**

The uncertainties will be added as filled color. We also estimate the statistical difference (t-test) for summer and winter in each sectors. The values are added in the figure captions.

**Fig S3: Interesting chart. This reader sees France with neither strong need nor strong source? How then did France come up with 225 TWh (Figure 3) for export to Germany?**

Yes, based on our analysis, France can be self-sufficient, however, it transfers LNG from ES to DE. You can see the dark green of ES. Similarly, NL and BE-LU transfers gas from UK to DE.

**Fig S4: Date units in top panel not correct? In lower panel, readers learn that storage accumulates during EU summers but supply for that storage arrives during winter? Evidence for annual "flow mass balance" but not for seasonal or monthly? Need to see uncertainties around these graphs?**

We have corrected the date error in the top panel to ensure accurate time representation. Storage normally accumulates in summer and is released as supply in winter. In this figure, the "to storage" values are directly obtained from physical flow (raw data), while the storage supply from different sources is based on model outputs. We acknowledge that uncertainty is important, but adding uncertainty estimates to stacked area plots is challenging due to visual

complexity. Instead, this figure primarily serves to help readers understand the storage-supply cycle and the reduction in Russian winter supply from storage.

**Table S2: In this data, France shows significant (?) increase in transmission over within-country consumption. Most French export went to Germany? Because authors present data as ratios, readers can't, from this Table at least, determine amounts?**
This table is a supplemental of Figure S3 and Fig 3. The amount value transferred (for all countries) can be found in the transportation graph in Fig 3.

**Fig S5: From this graph, reader concludes that Nordstream disruptions had larger impact, in terms of price, than invasion? Time required for sanctions / interuptions to impact prices? Additional source(s) of uncertainty?**
Literately, yes. While the Nordstream disruption had a larger and more immediate impact on gas prices, it was ultimately a consequence of the invasion. The price fluctuations observed in Fig S5 reflect the cumulative effects of geopolitical events, including the initial invasion, sanctions, supply disruptions, and market responses over time. We are not conducting a detailed microeconomic analysis of gas price fluctuations. Instead, the purpose of Fig S5 is to help readers understand how gas prices evolved in response to key geopolitical events and how stabilization occurred as EU countries adapted to the crisis.

**Fig S6: Reader needs information about uncertainty bands around gas and electricity prices?**
The data source only report an aggregated HEPI index value for EU27 per month. For the 2nd and 3rd panel, we already add the range of different country groups.

**Fig S7: Arrow widths provide only weak indicators? This figure does not show significant export France to Germany?**
Those figures are binary transportations, while the net export-import amount must be evaluated by aggregating both direction (as shown in Fig 3). For example, there is a wide red arrow from DE to FR, which means the export from DE to FR significantly reduce. And there is a thinner green arrow from FR to DE indicating the increment export from FR to DE. Together, it makes a significant net export increment from FR to DE.

**Fig S8: Daily flows clearly show impact of invasion but, for some exchanges, with lags of up to 6 months? One can get some info about temporal uncertainties from this figure?**
In our study and this figure (directly obtained using EUGasNet), we primarily aim to highlight how the direction of gas transportation changed following the invasion. However, readers can analyze more detailed transmission patterns, explore different strategies adopted by EU countries, and analyze geopolitical influences using our dataset EUGasNet.

**Figs S9a, S9b: need author declaration of pre-(orange?) and post-(blue?). Why does Portugal data show two separate clusters, unlike (for example) Spain?**
We have added text to caption to explicitly differentiate the pre-invasion (orange) and post-invasion (blue) periods in Figs S9a and S9b.
Portugal's warmer temperatures may explain the two separate clusters since our analysis fixes the winter period and assumes heating is enabled for the entire duration. This assumption may not hold in Portugal, which cause the lower consumption groups.

**Fig S11: No uncertainties here? From panels shown in S10, reader has no confidence in values below 0.3, or 0.5?**
Our attribution model conducted at daily resolution, and the values shown in Fig S11 are aggregated results. The unit used in the number shown on the map is TWh, meaning that even a value of 0.3 TWh represents a significant amount of gas consumption.

**Fig S12: Again, no uncertainties? This reader concludes that e.g. Austria used high proportions of renewables prior to invasion and that those proportions did not change post-invasion?**
Uncertainties are not required in Fig S12, as the data is based on aggregated daily power generation reports for the pre- and post-invasion periods, with energy shares calculated directly from raw reported data rather than model outputs.

Regarding Austria, its primary electricity source is hydropower (over 60%), which remained largely unchanged. However, the share of gas in Austria's electricity mix did slightly decrease during the post-invasion winters, reflecting a shift away from fossil fuel dependence.

**Fig S13: Reader needs information about uncertainties represented by color bands?**
The color bands present the SD of EU27&UK. We have added it into the captions.

**Fig S14: Difficult but important figure? For this reader, only Germany, Italy, France and UK, perhaps with Spain and perhaps with Austria, showed significant changes over two winters? Everything else (all other countries) within noise?**
Not really. Most EU countries experienced a significant drop in power generation during the post-invasion winters. However, the visualization challenge arises because absolute power generation levels vary significantly across countries, making changes less visible for lower-generation countries. To improve clarity, we have updated the figure by adding an extra y-axis showing the percentage change for each country.

**Fig S15: Uncertainties? Reader finds this figures potentially useful but not without some indication of uncertainty?**
We present this figure to visualize the day-to-day difference (all country aggregated), it is not suitable to add uncertainties. It will make the figure unreadable for the readers.

This figure is designed to visualize day-to-day differences in gas consumption across all aggregated countries, rather than providing detailed uncertainty estimates. Including uncertainties in this context would make the figure unreadable.
To ensure readers understand this, we modify the figure caption with *"Day-to-Day Power Generation Comparisons Across Energy Sources"*.

**Table S4: Very useful! Should appear as Appendix, referenced on Zenodo and in section 6?**

We will add it as Appendix in Zenodo dataset.

**RC2**

The dataset and associated documentation generate a value added. However, some of the assumptions made are not very plausible and lack motivation. In addition, some methods used are non-standard and it is not clear why standard state-of-the-art techniques are not used. In combination, these issues make it difficult for me to endorse publication of the current manuscript and data. Below I describe the concerns in more detail, followed by some minor comments, mostly of editorial nature.

The analysis makes a very strong assumption that in the industry sector, gas is substituted with electrically powered heat pumps (p. 3 l. 75) which seems not to be grounded in actual data. In reality, many companies, in particular in the first year after the Russian invasion into Ukraine, switched to other fossil fuels such as oil and coal to avoid disruptions of supply and suffering from high cost of gas. Also, a lot of applications that natural gas is used for in industry require high temperature levels that cannot be provided by heat pumps.

The "empirical temperature-gas-consumption (TGC) curves" introduced in the Methods section (p. 3 l. 99) appears to miss the well-established concept of heating degree days (HDDs) that are typically used to correct heating energy consumption for climatic variations across different years (or other time periods). HDDs in addition to temperature variations also make use of a reference building, taking into account at which temperatures heating is actually needed.

I don't understand why results of the gas demand reduction (p.7, l. 206) are presented as an average across both winters, not separately per winter as the situation clearly changed. Gas prices were clearly much higher in the 2022/23 winter than in the 2023/24 winter and also adaptation measures such as increasing LNG import capacity had been scaled up to significantly higher levels by the 2nd winter. More generally, prices as an explanatory variable seem to be not adequately considered in the analysis

In the power sector analysis (p. 5 l. 169), increased electricity imports don't seem to be included as a substitute for domestic gas-powered electricity generation. Moreover, policy interventions across different EU27 countries and the UK were quite different, leading to different incentives for power producers to substitute gas with other fuels. For example, in Spain gas-fired power generation was taken out of the general merit order approach and producers were compensated via different instruments to reduce electricity price spikes.

We sincerely thank the reviewer for the thorough and insightful comments. The major comments focus on the following concerns: (1) the methodology for modeling temperature-gas consumption relationships, (2) the assumptions in the industrial sector, (3) the analysis of gas demand changes and price dynamics across the two winters, and (4) the electricity imports as substitutes for domestic gas-powered generation. We generally agree with the issues raised by the reviewer and provide detailed explanations, and responses to each specific question below, outlining how we have addressed these concerns in our revised manuscript.

**1. Modeling temperature-gas consumption relationships**

As the reviewer suggested, HDD-based models are commonly used tools for estimating heating energy consumption. HDD is particularly effective when the base temperature—reflecting heating behavior patterns—is well-established through historical data. However, the aim of this analysis in our study is not only understanding the consumption due to the climate variables but also the changes due to the heating behavior changes between the pre-and post-invasion winters. In our assumption and observation, the base temperatures might change rapidly. So we used temperature-gas-consumption (TGC) curves in our study, which can be equally effective and perform better to capture the heating behavior changes.

Also, it is important to note that during winter periods, when most days require heating, both HDD-based models and TGC curves produce very similar outcomes. This is because HDD is essentially proportional to the extent to which the average daily temperature falls below the base temperature. In such cases, linear models based on either approach may yield comparable results for estimating the gas consumption vs temperature.

However, a key limitation of HDD-based models is the need to predefine a fixed base temperature. This assumption may not hold in the context of the significant changes in gas supply during the post-invasion winters. Our findings indicate that people tended to use less gas for heating at the same ambient air temperatures, suggesting a lower base temperature. This change is clearly depicted in the TGC curves in Fig. S9. The TGC approach allows us to accurately model energy consumption across different temperatures while simultaneously capturing shifts in heating behavior over the two periods.

In our revised manuscript, we have added the following sentence to **2.5.1 Residential heating sector** to clarify why HDD-based models were not used in this study:

*"We used Temperature-Gas Consumption (TGC) curves instead of Heating Degree Day (HDD) models, which are widely used in energy modeling, because TGC curves not only effectively model gas consumption during winter but also capture shifts in heating behavior between pre- and post-invasion winters. Unlike HDD models, which rely on a fixed base temperature, TGC curves account for changes in heating patterns, reflecting the observed reduction in heating demand at the same ambient temperatures during the energy crisis, as shown in Fig. S9."*

**2. Assumptions in the industrial sector**

As the reviewer pointed out, many industries adopted oil and coal as alternative fuels, particularly in the immediate aftermath of the energy crisis. Additional, certain industrial applications require high temperatures that heat pumps alone cannot provide. However, data limitations prevent a comprehensive analysis of fossil fuel substitution in the industrial sector.

To address this challenge, we used simplified assumptions to analyze a worst-case scenario: we assume that no alternative fossil fuels were available and evaluate the potential impact on industrial production if gas reduction were fully offset by electrically powered heat pumps. This scenario provides an upper-bound estimate of the potential industrial production loss due to gas shortages. If our results indicate that heat pumps could compensate for the decline in gas energy use, then even in this extreme case, industrial production would not be significantly affected. However, since we do not account for coal and oil substitution, our analysis may overestimate the negative effects of the gas crisis on industrial activity.

Although our approach may overestimate the actual impact on industrial production, the findings also highlight the **additional electricity demand required for a transition to greener energy sources** in the industrial sector. Electrifying industrial heating, particularly through heat pumps, would necessitate a **substantial expansion of clean electricity supply**. This underscores the **critical need to scale up renewable energy generation and strengthen grid capacity** to ensure that industrial production remains stable while reducing reliance on fossil fuels. This analysis also serves as a practical demonstration of our dataset's ability to assess industrial resilience to the gas crisis. Our dataset and analysis results highlight the gap in clean energy capacity needed to maintain industrial production at pre-invasion levels.

In the revised manuscript, we have clarified our assumptions, explicitly discuss their limitations, and provide a more transparent rationale for our scenario-based approach. Future research should incorporate detailed industrial fuel-use data to refine these estimates and assess the feasibility of alternative fuel pathways beyond electrification:

In 2.5.3 Industrial sector"

*"…Our assumption and analysis might overestimate the impact of gas shortages on industrial production (see Uncertainty section below)."*

In 3. Uncertainties and bias:

*"In the industrial sector, our simplified assumption does not account for the substitution of gas with other fossil fuels as energy source, such as oil or coal, due to the lack of reliable data, even though these fuels were widely used by industries during the energy crisis to avoid disruptions. Additionally, many industrial processes require high temperatures that heat pumps alone cannot provide. As a result, our analysis likely overestimates the impact of the gas crisis on industrial production. However, it serves as a worst-case scenario, providing an upper-bound estimate of potential industrial production losses without considering alternative fossil fuels (e.g., oil or coal) and relying solely on electricity, which implies the additional electricity demand required for industrial electrification in the transition to greener energy sources."*

In 4.3.3. Industrial sector

*"Although the actual impact on the industrial sector might be overestimated due to our assumption of no fuel-switching to fossil fuels, our upper-bound analysis highlights the significant industrial impact for EU countries, which in turn implies the substantial electricity demand required if the EU transitions toward industrial electrification to replace natural gas and other fossil fuels. While industrial heat pumps provide viable pathways for reducing reliance on fossil fuels, the large electricity demand suggested by our study would necessitate a significant expansion of clean electricity generation and grid capacity to further ensure the energy security in EU. The feasibility of this transition depends on the EU's ability to scale up renewable energy sources while reinforcing grid infrastructure to support increased industrial electricity consumption. "*

**3. Gas demand changes and price dynamics**

Separating the analysis for the two post-invasion winters could provide deeper insights into sector-specific dynamics, but we chose not to conduct a separate analysis because overall gas supply and consumption patterns remained largely consistent between the first and second post-invasion winters (Fig 2). Following the major structural supply shifts between the pre-invasion and first post-invasion winters, which happened rapidly — LNG replacing Russian pipeline imports and intra-EU gas redistribution stabilizing supply shortages—the trends remained stable into the second winter, with no significant shifts in total gas demand or supply sources. To further illustrate this point, we will add a new figure to the Supporting Information showing the day-to-day difference between the two winters.

Despite this general stability, we acknowledge that the power sector exhibited structural changes between the two post-invasion winters. Notably, Germany initially relied on coal for power generation but significantly increased its use of renewable energy in the second winter (Fig S14). This transition is already discussed in:

*5.5. Structural independent from gas-powered electricity*

*"In the initial post-invasion period, fossil fuel substitution remained significant, particularly in Germany and Italy (Fig S14a), accounting for 48.1% of the substitution in these two countries. However, by the second post-invasion winter (2023–2024), renewable energy took the lead across all EU27&UK countries, with renewables accounting for an increased substitution rate of 114.2%, suggesting that the structural shift away from gas-powered electricity has been successfully developing in the EU."*

Regarding gas prices, we view them as a reflection of market panic and supply expectations rather than a direct predictor of demand. As shown in Fig S5, LNG and Dutch TTF prices spiked sharply at the onset of the first winter due to fears of shortages but declined as EU countries stabilized supply through measures such as France redirecting gas to Germany and Germany commissioning new LNG terminals. Prices then remained stable through the second winter, even though supply and demand patterns showed little variation between the two post-invasion winters. The lack of a clear correlation between gas prices and demand fluctuations supports our decision not to use gas prices as a primary predictor of consumption changes. We hope this clarification addresses the reviewer's concerns and reinforces the robustness of our methodology and conclusions.

**4. Electricity imports**

The reviewer points out that our analysis does not explicitly include cross-border electricity imports as a substitute for domestic gas-fired generation and lacks a discussion of policy interventions across the EU27&UK. We agree that this limitation might lead to an overestimation of potential electricity shortages caused by the gas crisis.

However, we believe that cross-border electricity imports played a limited role in mitigating the crisis due to widespread power generation deficits across most EU countries. As shown in Fig S14, electricity generation declined in many countries compared to pre-invasion winters, reducing their ability to export surplus electricity. When multiple countries face simultaneous energy shortages, cross-border electricity trade becomes constrained, as there is insufficient surplus generation to meet regional demand. Among the few exceptions, Spain, Austria, and Poland exhibited some potential for electricity exports, largely due to increased renewable energy generation.

Despite these findings, only Spain played a crucial role in alleviating the EU gas crisis, as it was the only country with the potential to export both LNG and electricity. In the first post-invasion winter, Spain significantly increased renewable electricity generation while reducing its gas-fired electricity output only slightly, allowing it to export electricity to neighboring countries. In the second winter, Spain substantially reduced its gas consumption, might due to the removal of gas-fired power generation from the general merit order approach as suggested by the reviewer. This suggest that it could redirect LNG supplies to nearby countries to ease their gas shortages. This shift is reflected in changes in intra-EU gas transmission patterns (Fig 3), highlighting Spain's growing role as both an electricity and LNG supplier during the crisis.

To address the reviewer's concerns, we have explicitly highlighted Spain's role in alleviating the gas crisis, emphasizing how its renewable energy expansion and gas transmission strategies contributed to regional resilience:

4.3.2. Power sector

*"While cross-border electricity imports could theoretically mitigate some gas-related electricity shortages, their role was limited during the crisis due to widespread power generation deficits across most EU countries (Fig S14). However, Spain played a crucial role by significantly increasing renewable electricity generation in the first winter and reducing its gas consumption in the second winter, allowing it to support regional energy stability through both electricity exports and LNG redistribution (Fig 3)."*

**Minor comments:**

- **The precision with which some of the results are reported appear to exceed reasonable expectations regarding their uncertainty (e.g., 87.8% or 976.8 TWh per winter – p. 1 l. 22).**

  The numbers reported in **line 22** are derived from physical flow data provided by ENTSOG. Reporting three significant figures in this instance is appropriate because the

raw data are recorded in kWh at daily resolution and subsequently aggregated to TWh. This level of precision reflects the high resolution and reliability of the raw data.

However, we acknowledge that model-based results require careful consideration of uncertainties. Two significant figures are more appropriate in this situation to better reflect the inherent uncertainties in model-derived outputs. We have reduced the significant figures to two from model outputs throughout the manuscript.

- **While not wrong, some of the units used are not those commonly used in energy research (e.g., 10^9 m3 instead of bcm = billion cubic meters – p.2 l. 38).**

  We fixed units and number formatting, *"155×10^9 m3" → "155 bcm"*

- **The GHG balance of pipeline and LNG transportation of natural gas (p. 10 l. 344) depends on many factors, but importantly on the transport distance. Leakage aside, on shorter distances pipeline transport tends to come with lower emissions but with increasing transport distance the energy use of compressor stations increases proportionally to the distance while the biggest chunk of emissions by LNG tanker originates from the liquefaction process.**

  Thank you for your comment. Indeed, the GHG emissions from pipeline and LNG transportation systems can vary significantly due to various factors, including transport distance. Our argument focuses on the substantial increase in GHG emissions associated with switching from Russian pipeline gas to LNG.

  To support this, we compare the transportation system leakage percentage reported for the Russian pipeline (1.4%) with that of LNG transportation. Even when excluding emissions from the liquefaction process, LNG already demonstrates significantly higher GHG emissions. To further clarify this point, we have added a reference discussing the life cycle GHG emissions of LNG transportation and end use (Abrahams et al., 2015).

  The text has been revised as follows:

  "*Despite the large GHG emissions associated with LNG liquefaction, our estimations suggest that, solely during transportation, LNG tankers might produce 2.4 times the amount of CO2-equivalent emissions compared to pipelines when supplying the same amount of gas, even after accounting for a potential leakage rate of 1.4% from Russian pipeline transportation (see supplementary materials) (Lelieveld et al., 2005) (Abrahams et al., 2015).*"

- **The writing is somewhat repetitive and could benefit from being thoroughly edited.**

  We will revise the manuscript throughout, reduce the repetitiveness and improve clarity through editing.

**RC3**

**General Response**
Thank you for your encouraging comments and constructive suggestions. We appreciate your recognition of the manuscript's strengths and your detailed notes for improvements.

**Line 22 – Since these are based on estimates, 3 significant figures seem too precise given the modeling uncertainties. Suggest reporting all metrics to no more than 2 significant figures here and throughout the paper. In addition, any result based on models or estimates need to have uncertainties reported (e.g., 1 standard deviation or 95% confidence intervals).**

We appreciate your note regarding significant figures and uncertainty reporting. The values reported in line 22 are derived from physical flow data provided by ENTSOG, recorded in kWh at a daily resolution, and aggregated into TWh in our dataset. Given this direct measurement, reporting three significant figures for these values remains appropriate.

However, we agree that model-based results require careful consideration of uncertainties. Two significant figures are more appropriate in this situation to better reflect the inherent uncertainties in model-derived outputs. We have reduced the significant figures to two from model outputs throughout the manuscript. Additionally, we now consistently report uncertainty measures (e.g., standard deviations or 95% confidence intervals) for all model-based results.

**Line 23 – regarding LNG imports, would be helpful to name the countries dominating the increase in LNG exports to the EU.**

We agree that identifying the countries driving the increase in LNG exports to the EU would be especially important. However, ENTSOG data does not provide LNG sources at a daily resolution, as LNG is often stored before being distributed into the network, making precise allocation challenging.

To address this limitation, we are currently conducting a separate study that tracks LNG tankers and identifies the LNG export sources to specific EU countries. This research, combined with our network simulation model, will enable detailed reporting of LNG supply sources in future publications. We will present this study as soon as possible.

**In line 295, when reporting the p-value, is significance established at $p < 0.05$? If so, this should be stated in the methods. Note that significance can also be established at $p < 0.01$.**

We clarify the significance threshold ($p < 0.05$) in Methods section **2.5.2 Power Sector**, at Line 171. The text now states:

"*To better understand the substitution of renewable energy, we analyze Pearson's correlation coefficient (r) between the increase in renewable power generation and the reduction in gas-powered electricity, with statistical significance set at $p < 0.05$.*"

**There are several places where the authors make statements without providing supporting statistics in the main text, which are needed for clarity. For example, in section 4.3.1, what is the mean winter temperature (line 265) and what is meant by "warmer temperature anomalies" (line 267-268). In addition, the study states that there has been an increase in the adoption of heat pumps in the EU (line 374), without providing supporting statistics on the magnitude of such an increase. Similarly, in lines 376-377 – by how much did electricity prices increase? What is meant by "lower start heating temperatures" in line 378?**

1. Mean Winter Temperature (line 265):

We will added a table in the Supporting Information showing the mean winter temperatures of EU countries during pre-invasion winters (2019–2022) and the post-invasion winter (2022–2023).

2. Warmer Temperature Anomalies (line 267–268):

The term refers to the average day-to-day temperature difference, indicating whether a particular day is warmer or colder than previous temperatures. To clarify it, we have added the text:

*"...the average day-to-day temperature difference between pre- and post-invasion winters"* in Line 269.

3. Heat Pump Adoption (line 374):

While comprehensive datasets on heat pump installations remain unavailable, we referenced general reports and included a discussion of this limitation. Suggestions for reliable datasets are welcome for further exploration. Also, this would be an interesting future study topic about how the heat pump adoption will change the EU's energy landscape.

4. Electricity Prices (lines 376–377):

Electricity prices, including those for capital cities and the household energy price index (HEPI), are shown in **Fig. S6**. A significant price peak is visible in October 2022. Since detailed data on heat pump installations is unavailable, it is challenging to estimate the numerical impact of heat pumps on electricity prices.

5. Lower Start Heating Temperatures (line 378):

Start heating temperatures refer to the temperatures at which residential or public buildings activate heating systems. This concept aligns with the base temperature used in the Heating Degree Days (HDDs) model, which was suggested by another reviewer. However, we chose not to use HDDs in our analysis, as we hypothesize that the base temperature will be changed due to the change of heating behavior patterns between the pre- and post-invasion winters. This hypothesis is supported by **Fig. S9**, which shows that the Temperature-Gas-Consumption (TGC) curves for the post-invasion period (orange lines)

are generally lower than those for the pre-invasion period (blue lines). This shift indicates a reduction in the start heating temperature (or base temperature), reflecting changes in heating behavior. We added *"…, corresponding to the base temperature in Heating Degree Days (HDDs) model"* to clarify it.

**There are several typos in the manuscript that should be corrected (e.g., line 38, 318)**

Line 38:

Fixed units and number formatting:

*"155×10^9 m3" → "155 bcm"*

*"1.6×10^3" → "1.6×10$^3$"*

Lines 317–319:

We carefully reviewed these lines:

"*Using a gas-electricity conversion efficiency of 0.7, our reduction attribution model indicates that on 69.5% of the post-invasion winter days (Fig. 5a and 5b), the reductions in industrial gas consumption cannot be explained by the increases in total electricity generation.*"

No typos were identified in this section.

**Figure 5 (a). Font sizes in the figure legend and too small.**

We will increase the font size in Figure 5(a) to improve readability.